# Optogenetic induction of subcellular Ca²⁺ events in megakaryocytes and platelets using a highly Ca²⁺-conductive channelrhodopsin
Yujing Zhang[1], Jing Yu-Strzelczyk[2], Dmitri Sisario[1], Rebecca Holzapfel[1], Zoltan Nagy[1], Congfeng Xu [3], Chengxing Shen[3], Georg Nagel [2], Shiqiang Gao [2] ✉ & Markus Bender [1] ✉

Calcium signaling is crucial across various cell types, but its spatiotemporal dynamics remain difficult to study due to limited methods. Optogenetics, with its high precision, can address this challenge. In this study, we introduced the channelrhodopsin variant ChR2 XXM2.0, which exhibits high light sensitivity and enhanced Ca²⁺ conductance in *Xenopus* oocytes, into bone marrow-derived megakaryocytes through viral transduction, aiming to clarify the poorly understood role of Ca²⁺ dynamics in these cells. ChR2 XXM2.0 expression was confirmed in megakaryocyte membranes, and its functionality validated through whole-cell patch-clamp and calcium imaging. Localized activation of ChR2 XXM2.0 at the cell periphery induced cell polarization, dependent on localized calcium influx, myosin IIA, and integrin αIIbβ3-fibrinogen interaction. Furthermore, we generated a transgenic mouse line with *Pf4-Cre*-dependent expression of ChR2 XXM2.0, enabling optogenetic manipulation of anucleate blood platelets via light-triggered calcium signaling. Illumination induced phosphatidylserine and P-selectin exposure in spread platelets. Our results highlight the importance of asymmetric subcellular calcium events in megakaryocyte polarity and demonstrate the feasibility of manipulating platelet function using optogenetics. Taken together, our study introduces the ChR2 XXM2.0 construct and its corresponding Cre-dependent transgenic mouse line as powerful tools for manipulating subcellular Ca²⁺ signaling, with potential applications for different cell types.

The application of optogenetics originated in neuroscience and has transformed the field over the past decades[1]. Recently, it has expanded to other areas, such as plant biology and the cardiovascular system[2,3]. The popular optogenetic tool, Channelrhodopsin 2 (ChR2), was discovered in the green alga *Chlamydomonas reinhardtii* as a light-gated cation channel mediating phototaxis behavior[4,5]. It has been successfully utilized for the manipulation of membrane potential in neurons at the millisecond temporal level[4,6,7] and a red-shifted Channelrhodopsin (ChR), ChrimsonR, has been clinically applied in a blind patient for partial recovery of visual function[8]. However, ChR2 is a non-selective cationic channel protein, limiting its application when ion selectivity is crucial. Calcium (Ca²⁺) plays a pivotal role in many fundamental cellular processes across virtually all cell types. To explore the spatiotemporal dynamics of Ca²⁺ signaling in cell function, a wide range of

ChR variants (ChR2 H134R[4], CatCh[9], ChR2 XXM[10], PsCatCh 2.0[11] and CapChR2[12,13]) has been generated with improved Ca²⁺ conductance. In our previous study, XXM was compared with CatCh and demonstrated a higher Ca²⁺ current[10]. While PsCatCh 2.0 has advantages such as fast kinetics and high Ca²⁺ conductance, it also exhibits highly increased Na⁺ permeability[14]. Notably, we engineered a channelrhodopsin variant, ChR2 XXM2.0, from the ChR2 XXM[10] construct, introducing a further H134Q mutation and optimizing plasma membrane-targeted expression by addition of signal peptides and N-terminal truncation[15]. The ChR2 XXM2.0 was recently proven to be a powerful tool to study Ca²⁺ signaling in plant cells[15].

In this study, we demonstrate that ChR2 XXM2.0 expressed in *Xenopus* oocytes exhibits high light sensitivity and significantly higher Ca²⁺-current compared to ChR2 H134R and several other Ca²⁺-permeable ChR

[1]Institute of Experimental Biomedicine–Chair I, University Hospital Würzburg, Würzburg, Germany. [2]Department of Neurophysiology, Institute of Physiology, University of Würzburg, Würzburg, Germany. [3]Department of Cardiology, Sixth People's Hospital, Shanghai Jiaotong University School of Medicine, Shanghai, China. ✉e-mail: gao.shiqiang@uni-wuerzburg.de; bender_m1@ukw.de

variants, highlighting its potential as a tool to investigate spatiotemporal $Ca^{2+}$ signaling. Increasing evidence suggests that intracellular $Ca^{2+}$ influx in bone marrow megakaryocytes (MKs) contributes to thrombopoiesis[16–18], yet its precise spatiotemporal role in this process remains incompletely understood. We expressed ChR2 XXM2.0 in cultured bone marrow-derived MKs following viral infection and found that local $Ca^{2+}$ influx triggers polarized MK movement in vitro, suggesting a potential mechanism underlying MK polarization at sinusoidal blood vessels in vivo.

Transgenic mice engineered to express optogenetic tools in a cell type-specific manner offer a powerful approach for exploring the roles of molecules or proteins within particular cell types, especially anucleate cells, which are otherwise challenging to target. Among the existing ChR variants, only the ChR2 H134R variant with weak $Ca^{2+}$ conductance[4] has been used to generate a publicly available knock-in mouse line engineered for Cre-dependent expression of ChR2 H134R[19]. After validating the functionality of ChR2 XXM2.0 in MKs, we generated the Cre-dependent ChR2 XXM2.0 transgenic mouse line, enabling us to optogenetically target and manipulate anucleate blood platelets. $Ca^{2+}$ plays a vital role in platelet functions, and the sustained levels of supramaximal $Ca^{2+}$ is the prerequisite of procoagulant platelets[20]. The illumination of ChR2 XXM2.0-expressing platelets caused the exposure of phosphatidylserine and P-selectin, which are markers of procoagulant platelets. This work demonstrates, for the first time, the feasibility of optogenetic manipulation of anucleate blood platelets. ChR2 XXM2.0 proves to be a powerful tool for effectively regulating subcellular $Ca^{2+}$ dynamics in MKs, platelets, and potentially in other cell types.

## Results

### Comparison of different ChR2 variants for $Ca^{2+}$ conductance in oocytes

To choose an ideal tool for $Ca^{2+}$ modulation in primary cells, we compared $Ca^{2+}$ conductance across several high $Ca^{2}$-permeable ChR variants (Supplementary Fig. 1a), especially the recently published CapChR2[12,13] and also ChR2 H134R[4], since this ChR variant is widely used and a transgenic mouse line is available[19]. Our goal was to select a construct with both high $Ca^{2+}$ permeability and light sensitivity. Following equal cRNA injection into Xenopus oocytes, XXM2.0 exhibited the highest expression among all constructs (Supplementary Fig. 1b, c), a characteristic attributed to its LR, T, and E sequence modifications. To compare their $Ca^{2+}$ currents, we measured the photocurrents in the bath solution containing 80 mM $CaCl_2$, where the inward current is predominantly carried by $Ca^{2+}$. Before that, a final concentration of 10 mM $Ca^{2+}$ chelator BAPTA was injected into the Xenopus oocyte to avoid the activation of $Ca^{2+}$-activated $Cl^-$ channels (CaCCs). The ChR2 H134R[4] showed minor $Ca^{2+}$ conductance in comparison to ChR2 XXM[10], CapChR2[12,13] and ChR2 XXM2.0 (Fig. 1a, b). Notably, ChR2 XXM2.0 had the most significant $Ca^{2+}$ current compared with ChR2 H134R[4], XXM[10] and the recently published CapChR2[12,13], particularly under lower light intensities (Fig. 1a, b). This enhanced photocurrent likely results from a combination of increased expression levels (Supplementary Fig. S1), higher $Ca^{2+}$ selectivity[15], and potentially larger ion conductance. Instead of $Ca^{2+}$, further recordings were also performed with $Ba^{2+}$, which is the most similar cation to $Ca^{2+}$ but cannot activate the endogenous CaCCs. Similarly, ChR2 XXM2.0 showed the highest current with $Ba^{2+}$ (Fig. 1c). In comparison to ChR2 H134R[4], all three high $Ca^{2+}$-permeable ChR variants mutants showed prolonged off kinetics (Fig. 1d), indicating potential higher light-sensitivities[10].

### Functional expression of ChR2 XXM2.0 in mouse MKs

In blood platelets, the elevation of intracellular $Ca^{2+}$ contributes to various steps of cellular activation, but its role in MK differentiation and platelet production is less well defined. In order to enable spatially and temporally controlled manipulation of $Ca^{2+}$ signaling in MKs, we first characterized the expression and localization of ChR2 XXM2.0 in bone marrow-derived MKs after viral transduction. ChR2 XXM2.0, tagged with EYFP, co-localized with glycoprotein (GP) IX. This indicates that ChR2 XXM2.0 is expressed in both the plasma membrane and the internal demarcation membrane system

(DMS) of megakaryocytes (MKs) (Fig. 1e). The DMS provides a sufficient membrane supply for the formation of thousands of platelets from a single megakaryocyte. ChR2 XXM2.0 also partially co-localized with Orai1 at the plasma membrane and was detected to be in close proximity to Stim1, whereas the early endosomal marker EEA1 and lysosomal marker Lamp1 showed no co-localization with ChR2 XXM2.0 (Supplementary Fig. 2). To assess whether ChR2 XXM2.0 is functional in MKs, we applied whole-cell patch-clamp and measured the light-induced current in MKs. Upon illumination, a strong inward current indicating cation influx was detectable in ChR2 XXM2.0 MKs, whereas no photocurrent was detected in non-transduced MKs (Fig. 1f). We also analyzed MKs of the widely used optogenetic knock-in mouse line ChR2-EYFP with the mutation H134R[19] expressing Cre recombinase under the control of the platelet factor 4 (Pf4) promoter[21]. ChR2 H134R showed a similar expression pattern in MK membranes as compared to ChR2 XXM2.0 MKs (Supplementary Fig. 3a). However, the photocurrent in ChR2 H134R MKs was much weaker and was only ~8% of the ChR2 XXM2.0 MKs (Fig. 1f, Supplementary Fig. 3b). Long-term (90 min) global illumination of ChR2 XXM2.0 MKs resulted in phosphatidylserine (PS) exposure on the plasma membrane (Supplementary Fig. 4a, b). In contrast, the PS exposure percentage of ChR2 H134R MKs was reduced compared to ChR2 XXM2.0 MKs (~20% vs ~60%) (Supplementary Fig. 4c). Given the possibility of phototoxicity of the fluorescence protein, we only expressed YFP in MKs and globally illuminated MKs for 90 min. YFP-expressing MKs did not show an increased percentage of PS exposure (Supplementary Fig. 4d). This finding suggests that long-term channel opening is detrimental to cell. To confirm the $Ca^{2+}$ influx, we used Cal 590^{TM} to detect intracellular $Ca^{2+}$ dynamics in MKs during 3 min global illumination. Increased intracellular $Ca^{2+}$ levels were observable in ChR2 XXM2.0-positive MKs after illumination (Fig. 1g, h; Supplementary Video 1). Furthermore, spread ChR2 XXM2.0 MKs exhibited stress fiber formation and a further increase in spreading after illumination (Supplementary Fig. 5a–c). These data suggest that ChR2 XXM2.0 with enhanced $Ca^{2+}$ permeability is functional in bone marrow-derived MKs to trigger $Ca^{2+}$ influx.

### Local activation of ChR2 XXM2.0 induces MK polarization toward the illumination side

Mature bone marrow MKs are located next to sinusoidal blood vessels and extend long cytoplasmic protrusions, designated proplatelets, into the vessel lumen, from which platelets are released. MKs need to polarize as a prerequisite for directional proplatelet release into sinusoidal vessels. $Ca^{2+}$ has been reported to regulate cell polarity in plant cells and mammalian cells[22–24]. Therefore, we performed local illumination of ChR2 XXM2.0-positive MKs spread on fibrinogen (Fig. 2a). After 3 min local illumination in a peripheral region of MKs with blue light, ChR2 XXM2.0 MKs showed the capability of directional movement comparing to control MKs as determined by the distance between the center of mass of MKs before local illumination and at the end of the observation period (Fig. 2b–e; Supplementary Video 2). The majority of ChR2 XXM2.0 MKs showed a polarization behavior toward the illumination side, as evidenced by the polarization trajectories and the rose diagram (Fig. 2f, g). In contrast, ChR2 H134R MKs did not show polarized movement in response to local blue light (Supplementary Fig. 3c), suggesting a potential role of $Ca^{2+}$ in this process. To assess the role of $Ca^{2+}$ in light-induced polarization of ChR2 XXM2.0 MKs, spread MKs were preincubated with the cell-permeant $Ca^{2+}$ chelator BAPTA-AM or DMSO as control before local illumination. In the presence of 100 μM BAPTA-AM, the ChR2 XXM2.0 MKs showed impaired polarization compared with the DMSO-treated group (Fig. 2h, i, Supplementary Video 2). Taken together, these data demonstrated that local $Ca^{2+}$-influx is involved in polarized MK movement.

### The contractile protein non-muscle myosin IIA is involved in MK polarization

Non-muscle myosin IIA is an important regulator of adhesion and polarity in cell migration. Elevation of intracellular $Ca^{2+}$ has been shown to control

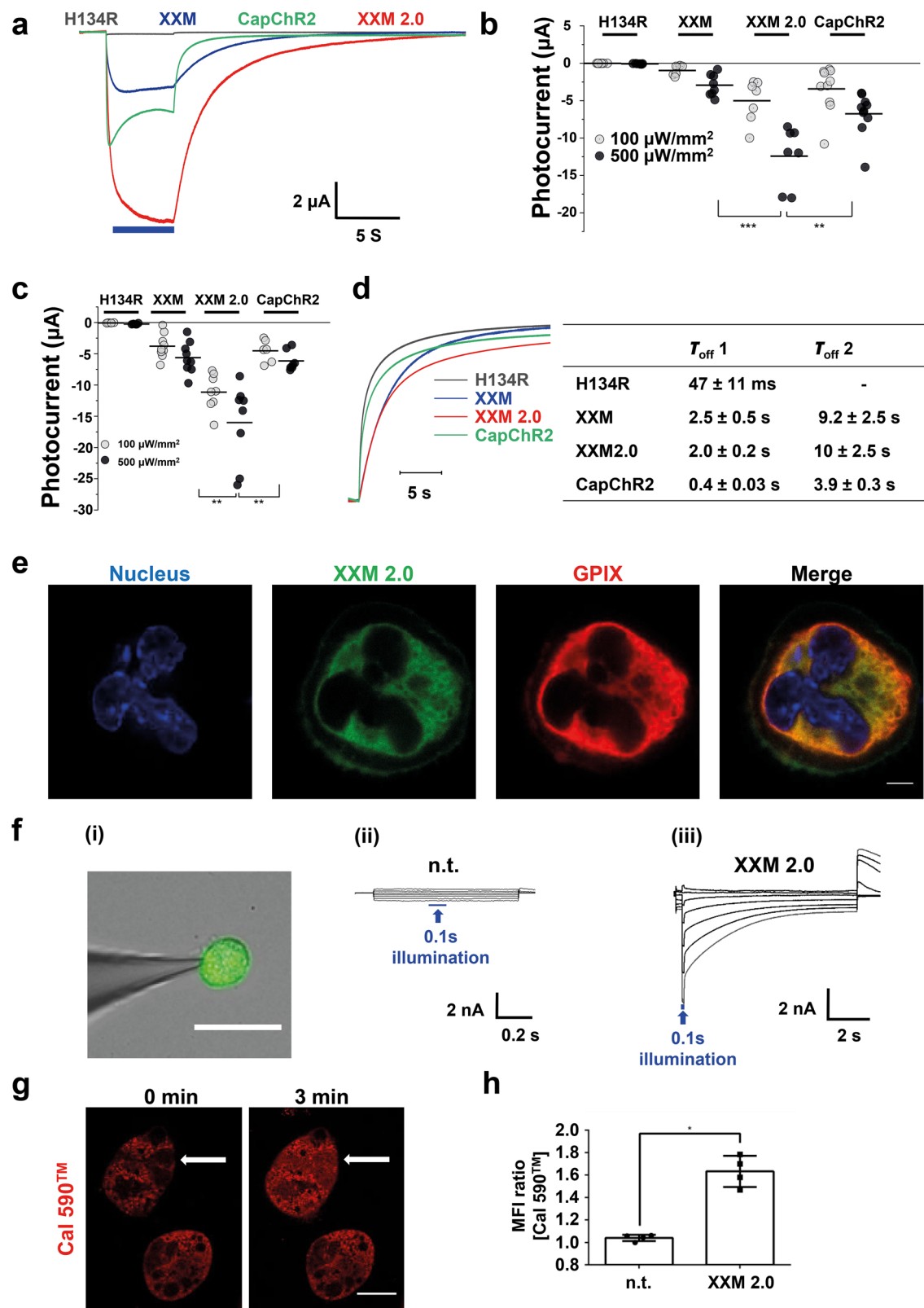

cell migration through activation of myosin IIa by generating contractile forces in fish epithelial keratocytes[25]. Furthermore, platelet migration was reported to be controlled by myosin IIA, which was activated by increased intracellular $Ca^{2+}$[26]. Thus, we hypothesized that light-induced MK polarization might also depend on non-muscle myosin IIA activity, which was supported by the increased myosin light chain 2 (MLC2) phosphorylation in

ChR2 XXM2.0 MKs after illumination (Fig. 3a). To further confirm this, spread MKs were preincubated with 100 μM blebbistatin to inhibit myosin IIA function. ChR2 XXM2.0 MKs showed impaired polarization in the presence of blebbistatin compared to DMSO-treated MKs (Fig. 3b, c), suggesting that myosin IIA-dependent force generation triggered by increased intracellular $Ca^{2+}$ is involved in light-induced ChR2 XXM2.0 MK

**Fig. 1 | Characterization of ChR2 XXM2.0 and comparison with other ChR2 variants. a** Representative photocurrent traces of different ChRs in oocytes at $-60$ mV after BAPTA injection in 80 mM $CaCl_2$, pH 9.0. Blue bar indicates light illumination of the 473 nm laser. **b** Comparison of calcium currents from different ChR variants in oocytes. ($n = 7$–$10$), two sample $T$-test, **$P < 0.01$, ***$P < 0.001$. **c** Comparing the photocurrents of different ChR variants in oocytes in 80 mM $BaCl_2$, pH 9.0. Recordings were performed at $-80$ mV. All data points were plotted in the figure, and the mean was indicated as the black line ($n = 6$–$8$). Two-sample $T$-test. **$P < 0.01$. **d** Representative recording traces of different ChRs in oocytes to show the off kinetics. Amplitudes of different ChRs were normalized to a similar size in order to visualize the direct comparison of kinetics ($n = 4$–$9$). Values for the fast ($\tau$off 1) or slow ($\tau$off 2) components of the biphasic off-kinetics are shown in the right table. **e** Representative confocal images of ChR2 XXM2.0 MKs stained for the nucleus (blue) with DAPI, GPIX (red) with Dylight 647 anti-GPIX antibody, and ChR2 XXM2.0 was visualized by its fusion protein EYFP (green). Scale bar: 5 µm. **f** (i) Image of whole-cell patch clamp of a ChR2 XXM2.0-positive MK. Superimposed current traces for non-transduced (ii) and ChR2 XXM2.0 expressing (iii) MKs after applying 0.1 s light pulse (473 nm, 550 µW/mm$^2$) from a series of voltage pulses from $-80$ to $+40$ mV of 1 s or 10 s duration with 20 mV increments from a holding potential of $-40$ mV. Scale bar: 100 µm. **g** Spread ChR2 XXM2.0 MK was pre-incubated with 10 µM Cal 590™ for 10 min before 3 min global illumination (488 nm) to visualize intracellular $Ca^{2+}$ changes. MK was spread on 100 µg/ml fibrinogen. White arrow indicates illuminated MK. Scale bar: 22 µm. **h** Quantification analysis of MFI (Mean Fluorescence Intensity) ratio of Cal 590™ (Fluorescence intensity at 3 min illumination/0 min illumination) using ImageJ. ($n = 4$). For MFI quantification, a freehand ROI tool was used to define the illuminated cell, creating the ROI. Subsequently, the MFI was calculated by subtracting the average background gray value from the mean gray value of the ROI. Mann–Whitney U test. *$P < 0.05$. n.t. indicates no transduction; XXM 2.0: ChR2 XXM2.0 expressing MKs. Results are mean ± s.d.

polarization. In addition, the illumination of ChR2 XXM2.0 MKs also induced increased phosphorylation of Src family kinase (SFK), Akt and extracellular signal-regulated kinase 1/2 (Erk1/2), indicating their potential involvement in light-induced ChR2 XXM2.0 MK polarization (Supplementary Fig. 6).

## Local activation of ChR2 XXM2.0 triggers localized binding of integrin αIIbβ3 to fibrinogen

As a major component of platelet signaling, intracellular $Ca^{2+}$ rise leads to platelet responses, among others, to activation of the integrin αIIbβ3, which is the dominant integrin on the platelet surface[27,28]. Thus, we sought to determine if the illumination of ChR2 XXM2.0 MKs results in activation of the fibrinogen receptor αIIbβ3. In order to observe integrin αIIbβ3 activation, fibrinogen labeled with Alexa Fluor 488 was added to the cell culture suspension, and a red fluorescence tag mKate2 was fused to ChR2 XXM2.0 instead of the EYFP. Subsequently, ChR2 XXM2.0-mKate2 MKs spread on fibrinogen were first globally illuminated (Fig. 4a). Illuminated MKs showed increased peripheral fibrinogen binding during the 10 min observation time, indicating activation of integrin αIIbβ3 (Fig. 4b, d). To assess the effect of polarized $Ca^{2+}$ influx on integrin αIIbβ3 activation, ChR2 XXM2.0 expressing MKs were locally illuminated at its peripheral region (Fig. 4a). Fibrinogen binding on the plasma membrane of MKs was detectable only in the illuminated area (Fig. 4c, e), indicating polarized integrin αIIbβ3 activation. Next, we addressed whether the polarized fibrinogen binding was due to changes in the distribution of the integrin αIIbβ3, and incubated the ChR2 XXM2.0-mKate2 MKs with the anti-integrin αIIbβ3 JON6-Fab Alexa Fluor 488 antibody. However, local illumination did not change the distribution of integrin αIIbβ3 (Supplementary Fig. 7a, b), suggesting that polarized $Ca^{2+}$ influx triggers local integrin activation rather than redistribution. To assess whether the binding between activated integrin αIIbβ3 and fibrinogen was necessary for light-induced MK polarization, spread MKs on fibrinogen were preincubated with 50 µg/ml JON/A-Fab antibody to block the activated integrin αIIbβ3 and prevent its binding to immobilized fibrinogen after local illumination. We found that blockade of the binding between integrin αIIbβ3 and fibrinogen inhibited light-induced polarized MK movement (Fig. 4f). These data demonstrated that binding of integrin αIIbβ3 to fibrinogen is important for light-induced MK polarization.

## The small Rho GTPase Cdc42 is crucial for light-induced MK polarization

In cells other than MKs, previous studies have shown that integrin binding to extracellular matrix proteins triggers selective activation of Rho GTPases, such as Cdc42[29,30], which induces cell polarization and migration[31,32]. Importantly, Cdc42 was reported to be involved in MK polarization and together with RhoA are molecular checkpoints that control transendothelial platelet biogenesis[33,34]. To test the role of the small Rho GTPases in polarized MK movement, we used an inhibitor-based approach and genetic knockout

mice. Our data show that cell polarization after local illumination of ChR2 XXM2.0 MKs is dependent on the small GTPase Cdc42, based on experiments using the Cdc42-specific inhibitor CASIN (Fig. 5a, b) and Cdc42-deficient MKs (Fig. 5c). We next studied the role of other small GTPases, such as RhoA, RhoB, and Rac1, in light-induced ChR2 XXM2.0 MK polarization. After preincubation with 30 µM of the Rho inhibitor Rhosin, ChR2 XXM2.0 MKs did not show impaired polarization (Fig. 5d), indicating RhoA and RhoB are not involved in light-induced ChR2 XXM2.0 MK polarization. Consistently, there was also no difference in light-induced ChR2 XXM2.0 MK polarization between wild-type MKs and $RhoA^{-/-}$ or $RhoB^{-/-}$ MKs (Fig. 5e). Furthermore, $Rac1^{-/-}$ MKs exhibited similar light-induced ChR2 XXM2.0 MK polarization compared to wild-type MKs (Fig. 5e). Taken together, we demonstrated that Cdc42, but not RhoA, RhoB, and Rac1, is crucial for light-induced ChR2 XXM2.0 MK polarization and motility mediated by $Ca^{2+}$.

## Local light activation of ChR2 XXM2.0 induces DMS polarization in MKs

DMS polarization is a prerequisite for directional proplatelet formation into sinusoidal vessels. Therefore, we investigated whether light-induced $Ca^{2+}$ influx also regulates DMS polarization in MKs. As ChR2 XXM2.0 also localizes in the DMS, we utilized the red fluorescence tag mKate2 of ChR2 XXM2.0-mKate2 as a marker to observe DMS arrangement and to avoid activation of ChR2 XXM2.0 during the observation period (Fig. 6a). We found ChR2 XXM2.0-mKate2, indicative for the DMS, polarized toward the leading edge of polarized MKs (Fig. 6a, b; Supplementary Video 3). In order to distinguish the DMS polarization from the cell movement, spread MKs were incubated with fibrinogen in suspension before local illumination to block polarized MK movement (Fig. 6c). While the polarized movement of ChR2 XXM2.0-mKate2 MK was prevented, the DMS still polarized toward the illumination area (Fig. 6d–g; Supplementary Video 3). This observation indicates that restructuring of the internal DMS is triggered by local $Ca^{2+}$ influx.

## Generation of mice specifically expressing ChR2 XXM2.0 in MKs and platelets

So far, the viral approach has only allowed us to study light-induced effects on MK, not platelet function. Despite progress over the years, producing platelets under in vitro conditions remains very challenging. To establish optogenetics in platelets and manipulate platelet functions by light, we generated the optogenetic ChR2 XXM2.0 transgenic mouse line by CRISPR/Cas-mediated genome engineering. The CAG promoter-loxP-PGK-Neo-6*SV40 pA-loxP-Kozak-ChR2 XXM2.0-EYFP -rBG pA cassette was cloned into the ROSA26 locus to obtain conditional (loxP) ChR2 XXM2.0 mice, which were finally crossed with *Pf4-Cre* mice[21] to remove the stop codon and express ChR2 XXM2.0-EYFP specifically in MKs and platelets (Fig. 7a). Similar to the viral approach, ChR2 XXM2.0 was expressed in MKs of the transgenic mice (Fig. 7b). It showed high photocurrent upon global illumination (Fig. 7c) and polarized movement upon focal illumination

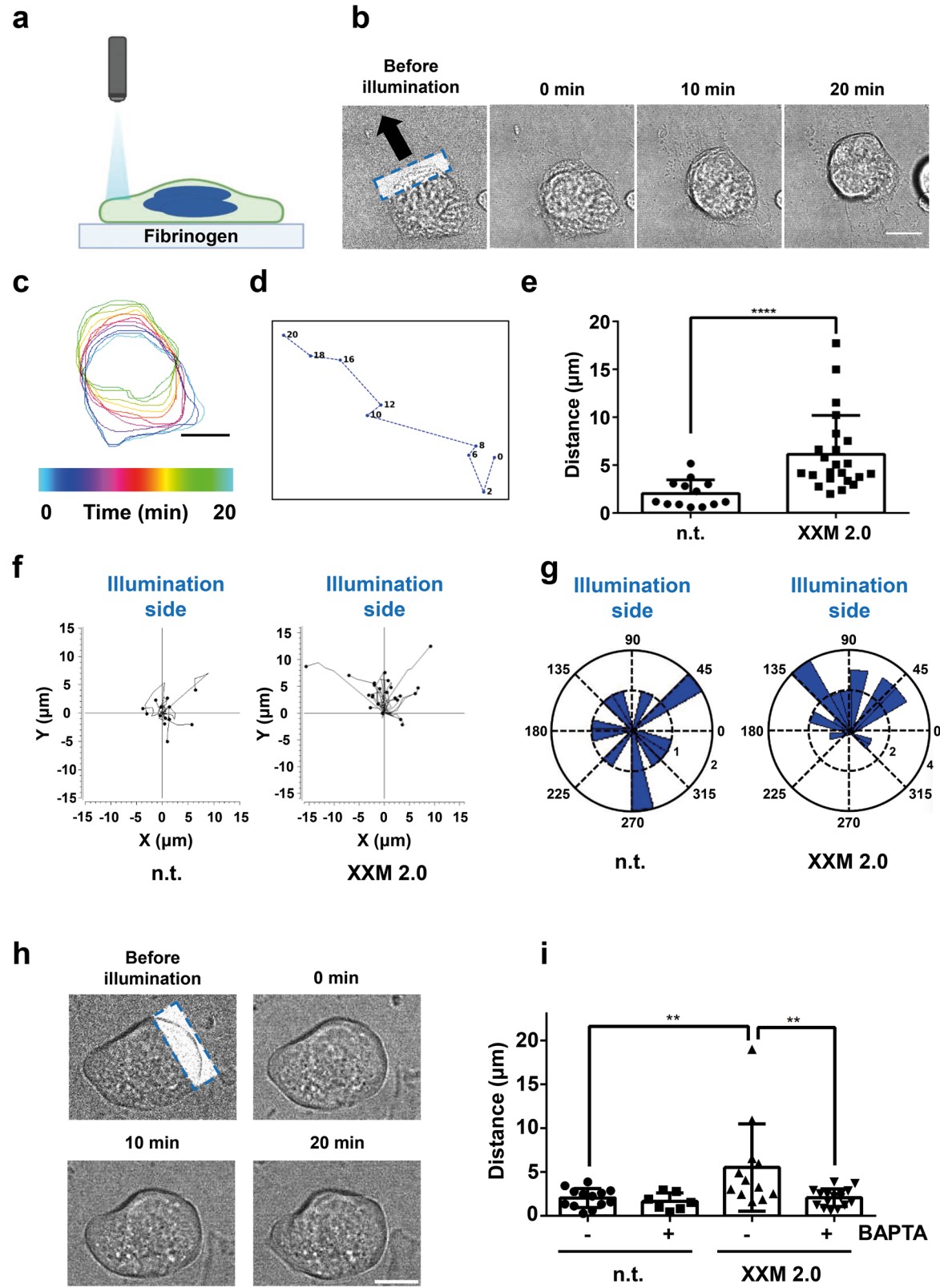

(Fig. 7d–f, Supplementary Video 4). Successful expression of ChR2 XXM2.0 was also confirmed in platelets of transgenic mice by immunoblotting and fluorescence microscopy (Supplementary Fig. 8a, b). PS exposure of platelets is induced by high sustained cytosolic $Ca^{2+}$ flux. This prompted us to test whether platelets from homozygous *ChR2 XXM2.0-EYFP*<sup>tg/tg, Pf4-cre</sup> mice spread on fibrinogen express PS after blue light exposure. Illumination of

half of the visual field resulted in a strong PS-positive signal followed by P-selectin exposure on the platelet membranes, while neighboring uni-lluminated platelets were unaffected (Fig. 8a–d, Supplementary Video 5). These results indicated that the ChR2 XXM2.0-EYFP platelets transition into a procoagulant state and highlighted the potential of optogenetics for manipulating the function of specific platelet subgroups within a

**Fig. 2 | Local illumination induces polarized movement of ChR2 XXM2.0 MKs. a** Schematic illustration of ChR2 XXM2.0 MK spread on 100 μg/ml fibrinogen and local illumination with blue light (488 nm). The schematic illustration was created with BioRender.com. **b** Representative images of time-lapse microscopy of ChR2 XXM2.0 expressing MK, which was locally illuminated for 3 min and then observed for 20 min. Local illumination was performed with FRAP module of a confocal microscope. The black arrow indicates polarization direction. The blue, dashed rectangle indicates illumination area. 0 min, 10 min, and 20 min mean the time point directly after 3 min local illumination. The scale bar is 22 μm. **c** Color-coded (time) representative cell outlines from the cell in (**b**) to show the polarization trajectory of MK after 3 min local illumination. Scale bar: 22 μm. **d** Dashed line represents the center of mass trajectory of the cell in (**b**), with numbers indicating observation time points after 3 min local illumination. **e** Distance of polarized movement of ChR2 XXM2.0 expressing MKs (XXM 2.0) and control MKs (n.t. indicates no transduction) post local illumination. The polarization distance was analyzed by the distance between the center of mass of MKs before 3 min local illumination and after 20 min

observation (n = 13–23). MKs from (**e**) are further analyzed in (**f**) and (**g**). Mann–Whitney U test. ****$P < 0.0001$. **f** Trajectories of MKs from after local illumination. **g** Rose diagram illustrates MK polarization direction after local illumination. The radius of the black dashed ring indicates cell count. The black dashed lines indicate orientation. **h** Representative images of time-lapse microscopy of ChR2 XXM2.0 expressing MK after preincubation with 20 μM BAPTA-AM, which was locally illuminated for 3 min and then observed for 20 min. Local illumination was performed with FRAP module of a confocal microscope. The blue, dashed rectangle indicates illumination area. 0 min means time point directly after 3 min illumination. The scale bar is 14 μm. **i** Distance of polarized MKs, which were preincubated with 20 μM BAPTA-AM or DMSO (-) and locally illuminated (n = 7–16). n.t. indicates no transduction; XXM 2.0: ChR2 XXM2.0 expressing MKs. Kruskal–Wallis test followed by Dunn's test for multiple comparisons. n.t. with DMSO versus XXM2.0 with DMSO, $P = 0.0072$; XXM2.0 with DMSO versus XXM2.0 with BAPTA, $P = 0.0055$. **$P < 0.01$. Results are mean ± s.d.

population. Taken together, the newly generated conditional ChR2 XXM2.0 mouse line (loxP/Pf4-cre) enabled precise manipulation of cellular $Ca^{2+}$ influx, thereby regulating MK and platelet function.

## Discussion

We expressed the ChR2 XXM2.0 in primary bone marrow-derived MKs and found that blue light illumination induced a significant photocurrent and $Ca^{2+}$ signals in those cells. Local illumination of MKs triggered polarized movement on fibrinogen, which was dependent on integrin αIIbβ3-fibrinogen binding, Cdc42 activity and myosin IIA activity. Further, we have established a new transgenic mouse line, which enables expression of ChR2 XXM2.0 in different cell types when combined with available Cre lines. Employing such an efficient transgenic expression strategy is especially crucial for cells like blood platelets, which would otherwise be unable to express the photosensitive protein. We capitalized on the *Pf4-Cre* driver line to express ChR2 XXM2.0 in platelets and found $Ca^{2+}$-dependent phosphatidylserine and P-selectin exposure of spread platelets upon illumination. These findings demonstrate the utility of this new optogenetic transgenic mouse model for manipulating $Ca^{2+}$ signaling and highlight its potential for broader applications.

With the help of ChR2 XXM2.0, we manipulated $Ca^{2+}$-triggered MK polarized movement at the subcellular level of a single cell for the first time. To our knowledge, this kind of manipulation has not been performed with a high $Ca^{2+}$-conductive ChR up to now. A similar manipulation was achieved by us in plant pollen tubes, which showed light-guided growth direction by activation of an anion-conducting channelrhodopsin *Gt*ACR1[35]. Recently, several publications used different ChR2 variants to induce cell migration[12,36,37], where they utilized the temporal precision of optogenetics to generate $Ca^{2+}$ oscillations and enhance the general motility strength at the whole-cell level. Our research reinforced another feature of optogenetics, the spatial precision, to dissect MK polarization at the subcellular level. Optogenetic manipulation of small GTPases also achieved similar light-regulated cell polarizations, e.g., cell protrusions and ruffling[38]. Differently, we targeted the second messenger $Ca^{2+}$ directly and established this method in MKs. Our successful approach suggests that ChR2 XXM2.0 can be useful in other cell types to decode the physiological meaning of subcellular $Ca^{2+}$ dynamics. Channelrhodopsins with high $Ca^{2+}$ conductance are highly desired for the spatially and temporally precise manipulation of $Ca^{2+}$ dynamics. Some non-channelrhodopsin-based light-gated $Ca^{2+}$ tools show limitations of dependence on cell type-specific endogenous proteins or very slow kinetics[39–42]. Recently, new $Ca^{2+}$-permeable channelrhodopsins (CapChR) were developed with enhanced $Ca^{2+}$ to $Na^+$ conductance ratio[12]. In comparison to CapChR2, ChR2 XXM2.0 showed a higher absolute $Ca^{2+}$ current in *Xenopus* oocytes. The ratio of $Ca^{2+}$ current to $Na^+$ current from CapChR2 was increased to ~2 in the buffer containing 70 mM $Ca^{2+}$ or 144 mM NaCl[12]. However, the $Na^+$ conductance can never be excluded for all the current $Ca^{2+}$-permeable ChR variants. But the electrochemical driving force for $Ca^{2+}$ to enter the cell is much higher than $Na^+$ under the

physiological conditions of mammalian cells. In our study, we also used the $Ca^{2+}$ chelator BAPTA-AM to confirm that the light-induced effect is from the $Ca^{2+}$ conductance of the ChR2 XXM2.0. In addition, the ChR2 H134R[4,19] with very weak $Ca^{2+}$ conductance was able to trigger MK membrane potential depolarization, but could not evoke MK polarized movement, again showing that the high $Ca^{2+}$ conductance is necessary for this process. Comparing to ChR2 H134R[4] and CapChR2[12,13], ChR2 XXM2.0 showed prolonged off kinetics. Fast kinetics are highly expected in excitable cells like neurons, but in non-excitable MKs, off kinetics in the second range was fast enough for our study. More important to MKs is the application of a mild light intensity to reduce the potential photo-damage from long-time illumination.

The molecular mechanisms that orchestrate the complex process of platelet biogenesis are still incompletely understood. To identify key proteins involved in MK differentiation and platelet production, most studies so far have relied on genetically modified mouse lines or chemical reagents. To our knowledge, a reversible approach to trigger intracellular processes and directly visualize cell behavior in a high spatiotemporal resolution has not been used in MK biology until now. In this study, we used the optogenetic approach to manipulate $Ca^{2+}$-signaling and directly observe MK behavior with high spatial resolution, and to shed new light on the role of $Ca^{2+}$ in megakaryo- and thrombopoiesis. Our in vitro findings point toward a potential role for local $Ca^{2+}$ rise in MK polarization toward the sinusoidal vessel, a prerequisite for platelet production. In support of these results, studies unveiled that deletion of the glutamate-gated N-methyl-D-aspartate receptor (NMDAR) with high $Ca^{2+}$ permeability in MKs results in reduced platelet counts in mice[18], and inhibition of extracellular $Ca^{2+}$ inflow also affected MK interaction with extracellular matrix proteins[17]. However, knockout mice of the major store-operated $Ca^{2+}$ entry (SOCE) components STIM1 and Orai1 mice displayed normal platelet counts[43,44]. This finding might be explained by a compensatory mechanism of different $Ca^{2+}$ channels in MKs. We may have observed increased MK mobility in our experimental system due to the lack of neighboring cells, which typically exist in the bone marrow. In light of this, our findings rather point to a role of local elevation of intracellular $Ca^{2+}$ in MK polarization than migration in vivo. What might cause polarized $Ca^{2+}$ influx in MKs and lead them to polarize toward blood vessels? Studies have shown that thrombopoietin (TPO) and stromal cell-derived factor 1 (SDF-1) can induce $Ca^{2+}$ influx in human MKs[16,45]. TPO is mainly produced by the liver and transported to the bone marrow through the bloodstream, while SDF-1 is primarily generated by endothelial and perivascular mesenchymal stromal cells[46,47]. The directed release of SDF-1 and TPO from the blood vessel toward the bone marrow might trigger polarized $Ca^{2+}$ influx in MKs, causing them to polarize toward the blood vessel. A central role in establishing cell polarity has been demonstrated for the small GTPase Cdc42 of the Rho family[32,48]. During the last years, experimental evidence has accumulated suggesting that Cdc42 activity in MKs is associated with polarized DMS formation and transendothelial platelet biogenesis[33,34]. Our results support these findings because

## a

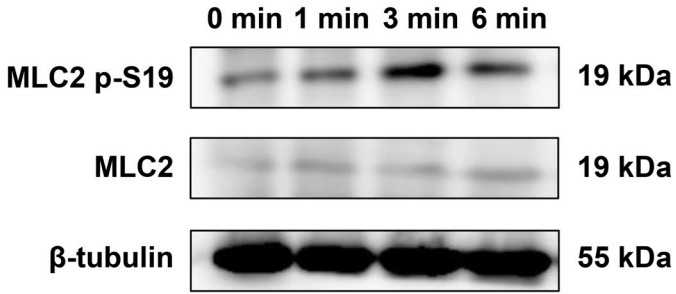

## b

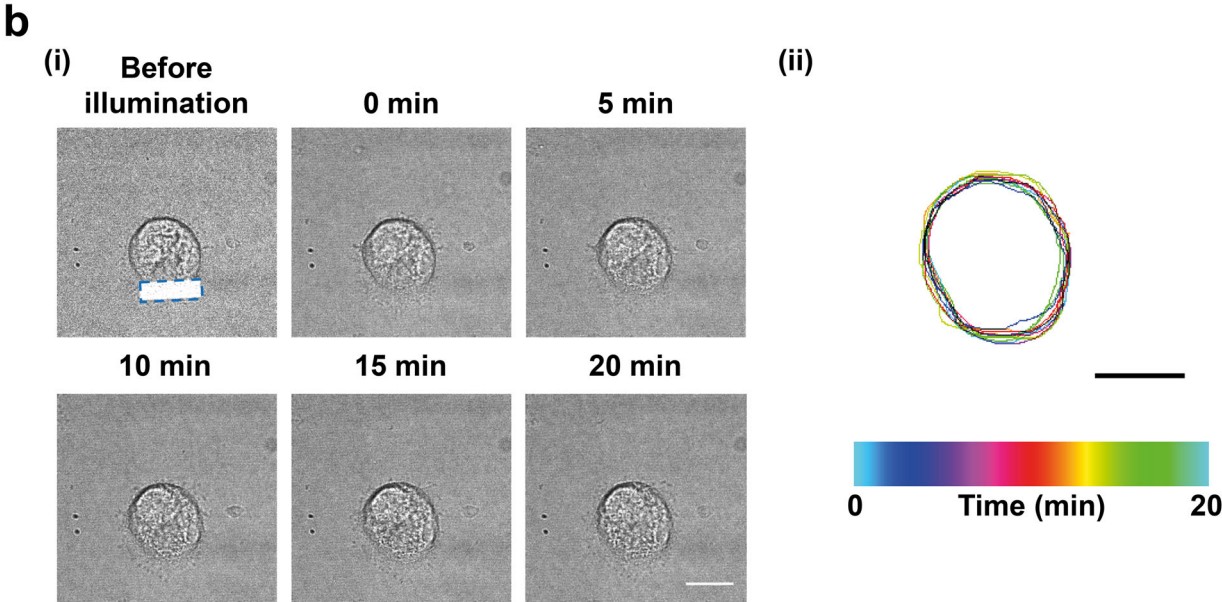

## c

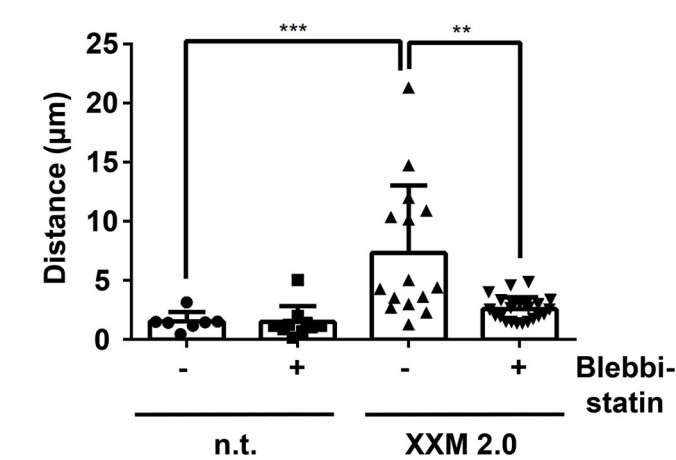

**Fig. 3 | Light-induced ChR2 XXM2.0 MK polarization depends on myosin IIA activity. a** Immunoblot analysis of phosphorylation of MLC2 at position S19 and total MLC2 expression after 0 min, 1 min, 3 min, 6 min global illumination of ChR2 XXM2.0 expressing MKs. **b** (i) Representative images of time-lapse microscopy of ChR2 XXM2.0 expressing MK, which was preincubated with 100 µM blebbistatin. The blue, dashed rectangle indicates illumination area. (ii) Color-coded (time) representative cell outlines from cell in (i). 0 min means time point directly after 3 min illumination. Scale bar: 22 µm. **c** Distance of MKs, which were preincubated with 100 µM Blebbistatin or DMSO (-) and then illuminated. n.t. indicates no transduction. XXM 2.0 indicates expression of ChR2 XXM2.0 ($n = 7$–22). Kruskal–Wallis test followed by Dunn's test for multiple comparisons. n.t. with DMSO versus XXM2.0 with DMSO, $P = 0.0004$; XXM2.0 with DMSO versus XXM2.0 with Blebbistatin, $P = 0.0095$. **$P < 0.01$, ***$P < 0.001$. Results are mean ± s.d.

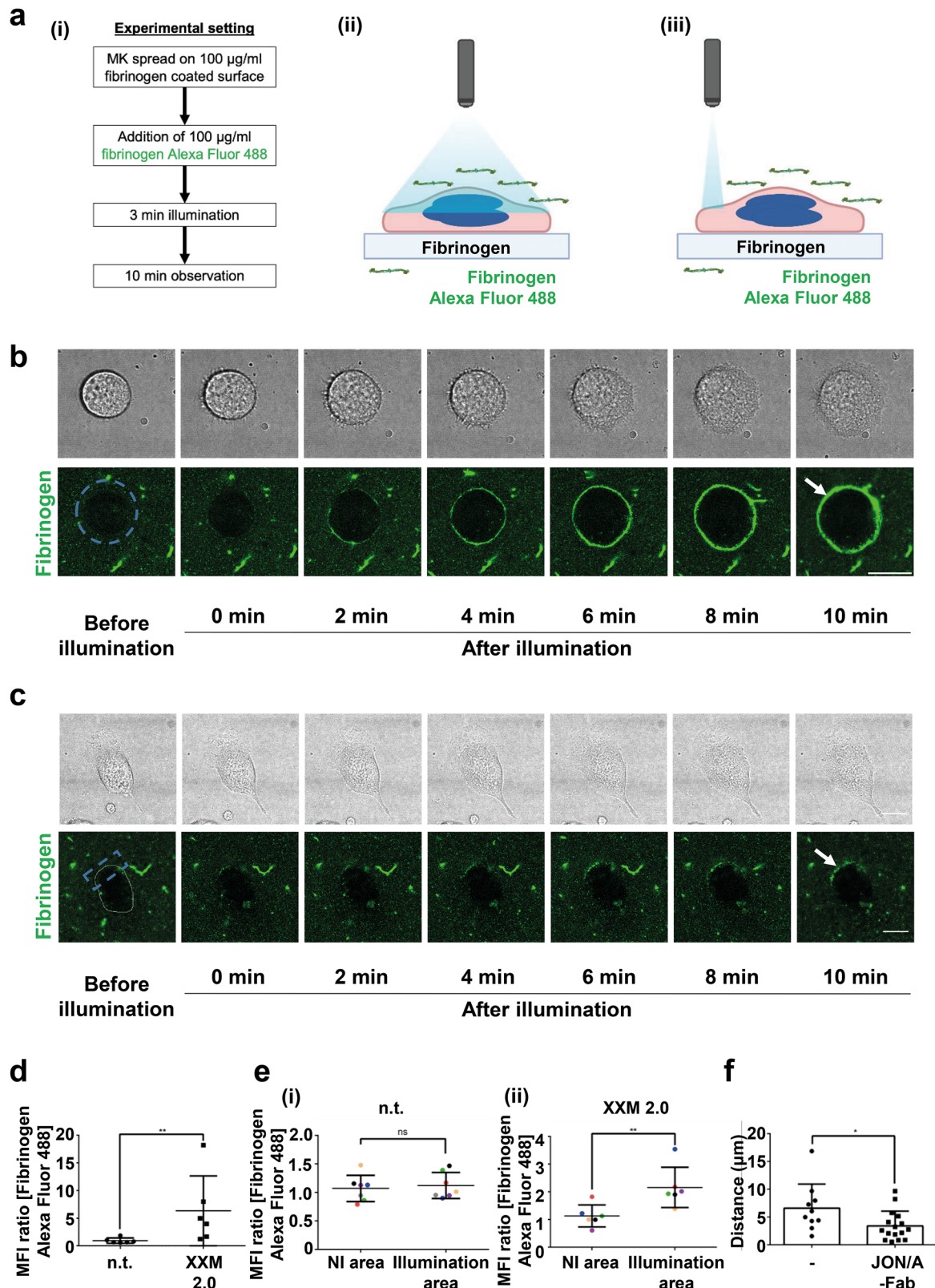

Cdc42 deficiency or inhibition of its activity prevented MKs from polarized movement. However, the detailed molecular mechanism of how Cdc42 is involved in this light-induced process has not been addressed. It was reported that the $Ca^{2+}$-dependent interaction of Lis1 with IQGAP1 promotes Cdc42 activation and induces neuronal motility[49]. Another study demonstrated that the interaction of integrins with the extracellular matrix

at the newly formed cell front leads to the activation and polarized recruitment of Cdc42 in astrocytes[30]. We demonstrate that the polarized $Ca^{2+}$ influx triggers polarized MK movement in an integrin αIIbβ3-dependent manner on a fibrinogen-coated surface. Of note, fibrinogen is localized in vascular sinusoids in the bone marrow[50], and it was described that both SDF-1 and TPO increase the adhesion of megakaryoblasts to

**Fig. 4 | ChR2 XXM2.0 MK binds to fibrinogen via αIIbβ3 at site of illumination.**
**a** Schematic of (i) experimental setting, (ii) global illumination and (iii) local illumination. The schematic illustration was created with BioRender.com.
**b, c** Representative time-lapse images of spread ChR2 XXM2.0 mKate2 MKs, which were incubated with 100 µg/ml fibrinogen Alexa Fluor 488 before 3 min **b** global illumination or **c** local illumination. 0 min means time point directly after illumination. The blue, dashed circle or rectangle indicates illumination region. White circles indicate the outline of cells. White arrows indicate the bound fibrinogen. MKs appear as black voids against the green fibrinogen backdrop. Scale bar: 22 µm.
**d, e** Quantification of MFI ratio of peripheral fibrinogen binding (Fluorescence intensity at 10 min post illumination/0 min) in MKs after **d** global illumination and **e** local illumination. MFI quantification involved delineating the illuminated and non-illuminated sections of the cell using a freehand ROI tool. This generated the ROI, and subsequently, the MFI for each area was determined by subtracting the average background gray value from the mean gray value of the ROI. The "illumination area" in (**e**) indicates the area of the cell that was illuminated, as indicated by the blue rectangle in (**c**), and the "NI area" in (**e**) indicates the area in the cell without illumination in (**c**). The same color of the data points of the NI area and the illumination area indicates the same cell. n.t. indicates no transduction; XXM 2.0: ChR2 XXM2.0 expressing MKs (**d**, $n = 6$; **e**, $n = 6$–7). **f** Distance of polarized movement after local illumination of spread ChR2 XXM2.0 MKs, which were incubated with 50 µg/µl JON/A-Fab to block the binding site of activated integrin αIIbβ3 to fibrinogen ($n = 10$–16). (-) indicates no incubation with JON/A-Fab. Statistics in (**d**–**f**): Mann–Whitney U test. $*P < 0.05$, $**P < 0.01$. Results are mean ± s.d.

fibrinogen through activation of integrin αIIbβ3[51]. Fibronectin is also a key matrix protein and ligand of αIIbβ3, which was reported to be important in αIIbβ3-mediated thrombosis and hemostasis[52]. However, we did not observe ChR2 XXM2.0 MK polarization following local illumination on a fibronectin-coated surface (Supplementary Fig. 9). The difference between fibrinogen- and fibronectin-dependent responses that we observed here is intriguing and deserves further in-depth investigation. One possible explanation might be that fibronectin can bind to αIIbβ3 but primarily engages α5β1, whereas fibrinogen is the main physiological ligand of αIIbβ3 and provides multiple high-affinity, multivalent binding motifs, such as the γ-chain HHLGGAKQAGDV sequence, which is absent in fibronectin[28]. Taken together, our findings suggest that the integrin αIIbβ3 is involved in MK polarization in vivo. Interestingly, β3-integrin-deficient mice display a moderately decreased platelet count[53,54]. However, patients with Glanzmann thrombasthenia (an inherited platelet disorder caused by mutations in the *ITGA2B* and *ITGB3* genes encoding the αIIbβ3 integrin) present with a normal platelet count[55], but can be at the lower range of normal[56]. Importantly, it has also been reported that rare gain-of-function mutations in those genes cause macrothrombocytopenia[57–62]. Consistent with our data, these observations point to a potential role of αIIbβ3 in platelet biogenesis.

The conversion of activated platelets to a procoagulant state involves specific biochemical and morphological changes, including PS exposure induced by prolonged elevation of cytosolic $Ca^{2+}$[20,63]. ChR2 XXM2.0 precisely induced PS and P-selectin exposure on spread platelets on a fibrinogen-coated surface upon illumination. This observation demonstrates the feasibility of manipulating platelet function with optogenetic tools. We propose that the conditional (loxP) ChR2 XXM2.0 mouse line can be a useful tool to study the spatiotemporal role of $Ca^{2+}$ influx in different cell types.

In summary, our study successfully implemented optogenetic manipulation of $Ca^{2+}$ signaling in mouse bone marrow MKs and platelets, and revealed that local $Ca^{2+}$ influx regulates MK polarization, pointing to a new mechanism of MK positioning toward sinusoidal vessels in vivo. The optogenetic tool ChR2 XXM2.0 and the newly developed transgenic mouse line hold promise for further investigations into subcellular $Ca^{2+}$ signaling dynamics across diverse cell types, both in vitro and in vivo.

## Methods
### Mice
Female and male mice used for experiments were at least 6 weeks old. To generate the conditional ChR2 XXM2.0-EYFP transgenic mice, the gRNA to mouse ROSA26 gene, the donor vector containing "CAG promoter-loxP-PGK-Neo-6*SV40 pA-loxP-Kozak-XXM 2.0-EYFP-rBG pA" cassette, and Cas9 mRNA were co-injected into fertilized mouse eggs to generate targeted conditional knock-in offspring by Cyagen Biosciences (Guangzhou) Inc. F0 founder animals were identified by PCR followed by sequence analysis, which were bred to wild-type mice to test germline transmission and F1 animal generation. B6.Cg-Gt(ROSA)26Sortm32(CAG-COP4*H134R/EYFP)Hze/J [stock number 024109: Ai32(RCL-ChR2(H134R)/EYFP)][19] transgenic mice were purchased from Jackson Laboratory. Transgenic mice (with loxP sites) were intercrossed with mice carrying the Cre-recombinase under the Pf4 promoter[21] to generate platelet- and MK-specific knockout mice. Mice are further referred to as ChR2 XXM2.0 (+ *Pf4-Cre*) and ChR2 H134R (+ *Pf4-Cre*), respectively. $Cdc42^{fl/fl Pf4-cre}$ mice ($Cdc42^{-/-}$)[64], $RhoA^{fl/fl Pf4-cre}$ mice ($RhoA^{-/-}$)[65], $Rac1^{-/-}$[66] and $RhoB^{-/-}$[67] were previously described. Bone marrow isolation was performed following cervical dislocation of mice under isoflurane anesthesia. We have complied with all relevant ethical regulations for animal use. The procedure for mouse bone marrow isolation was approved by the local government (Bezirksregierung Unterfranken).

### DNA plasmids and viral constructs
The ChR2 H134R and XXM plasmids for *Xenopus* oocyte expression were from the stock of the Nagel/Gao group. The CapChR2 was generated by quick change point mutations by Chong Zhang of the Nagel/Gao group with the CoChR L112C plasmid synthesized by GeneArt Strings DNA Fragments (Life Technologies, Thermo Fisher Scientific), according to the published amino acid sequences[13]. Plasmids for *Xenopus* oocyte expression were linearized by NheI digestion. cRNAs were generated by in vitro transcription with the AmpliCap-MaxT7 High Yield Message Maker Kit (Epicentre Biotechnologies), using the linearized plasmid DNA as templates. The ChR2 XXM2.0 was cloned into the murine stem cell virus (MSCV) vector between the restriction sites BamHI and HindIII. The ChR2 XXM2.0-mKate2 was developed by replacing the EYFP tag of ChR2 XXM2.0 with mKate2. The EYFP was amplified by PCR with BamHI and HindIII sites on the N and C terminal, respectively, and cloned into the MSCV vector by ligation after digestion with BamHI and HindIII. Plasmids newly generated in this study will be deposited to Addgene with sequence information and will be available online when the paper is published.

### Reagents and antibodies
Calcium 590™ (AAT Bioquest), DMSO, Rhosin (Merck), BAPTA-AM and 4',6-Diamidino-2-Phenylindole, dihydrochloride (DAPI) (Invitrogen), blebbistatin, Fibrinogen conjugated to Alexa Fluor 488 (Sigma), Fibronectin (Sigma), anti-MLC2 p-S19 (Sigma Aldrich), anti-MLC2 (Cell Signaling), CASIN (Sigma Aldrich), or anti-β-tubulin (Sigma Aldrich), anti-GFP and anti-GAPDH antibodies (Cell Signaling) were purchased. The antibodies JON/A (against activated form of integrin αIIbβ3, Emfret Analytics)[68], p0p6 (against GPIX, Emfret Analytics)[69], anti-P-selectin (Wug.E9, Emfret Analytics) and JON6-Fab Alexa Fluor 488 antibody (against integrin αIIbβ3, unpublished) were modified in our laboratory.

### Megakaryocyte culture
On day 0, bone marrow cells were obtained from femur and tibia of mice by flushing, and lineage depletion was performed using an antibody cocktail of anti-mouse CD3 (#100202, clone 17A2); anti-mouse Ly-6G/Ly-6C (#108402, clone RB6-8C5); anti-mouse CD11b (#101202, clone M1/70); anti-mouse CD45R/B220 (#103202, clone RA3-6B2); anti-mouse TER-119/Erythroid cells (#116202, clone Ter-119) (1.5 µg of each antibody per mouse, Biolegend) and magnetic beads (Dynabeads® Untouched Mouse CD4 Cells, Invitrogen). Lineage-negative (Lin-) cells were cultured in DMEM medium (supplemented with 10% FCS, 1% penicillin/streptomycin) containing 100 U/ml recombinant hirudin (Hyphen Biomed) and 1% TPO from supernatant (homemade) at 37 °C in 5% $CO_2$ overnight. A

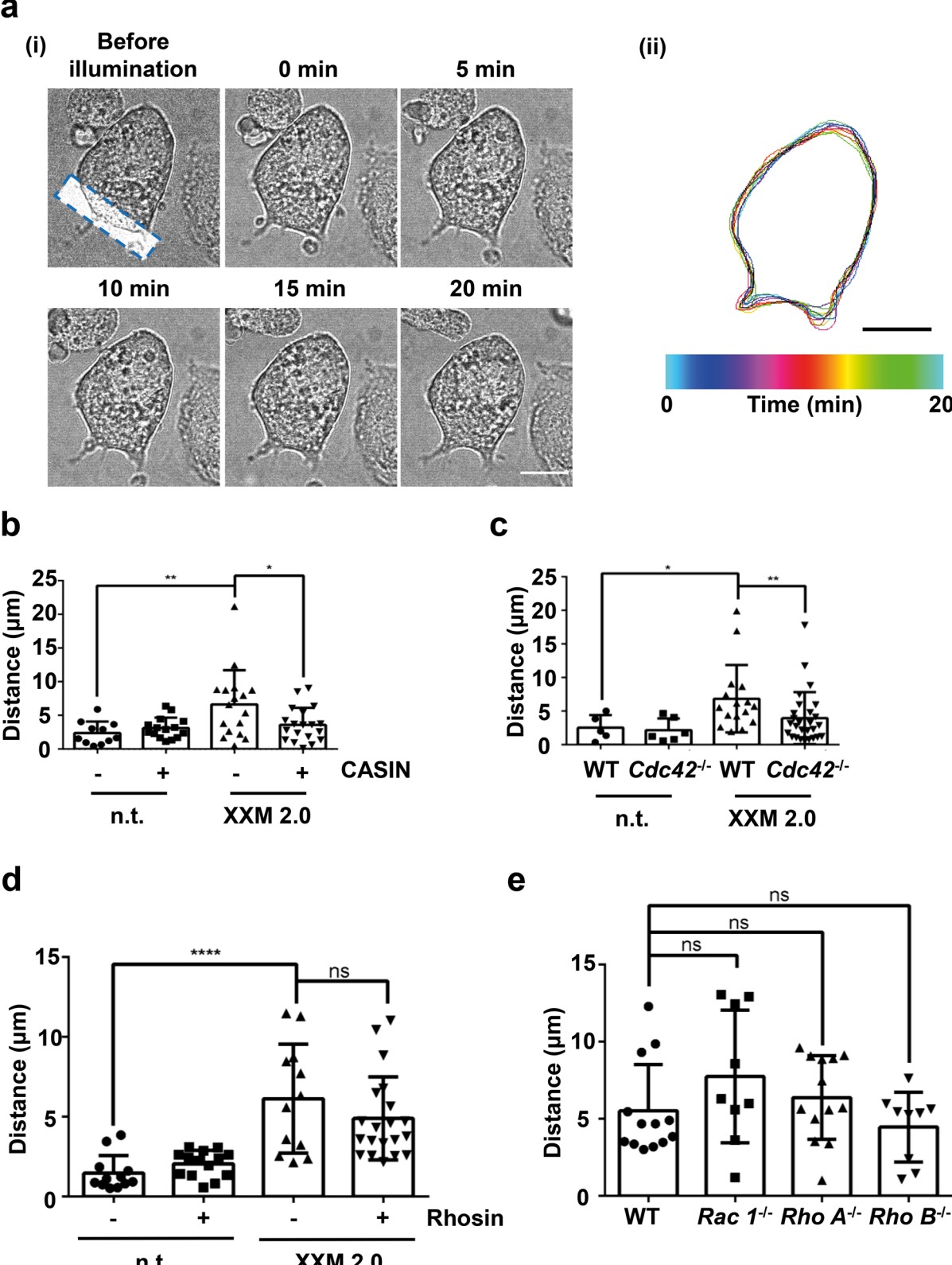

**Fig. 5 | Deficiency or inhibition of Cdc42 affects light-induced ChR2 XXM2.0 MK polarization.** a (i) Representative images of time-lapse microscopy of ChR2 XXM2.0 MK treated with 10 µM CASIN. The blue, dashed rectangle indicates illumination region. Scale bar: 22 µm. (ii) Color-coded (time) representative cell outlines from cell in (i). Scale bar: 22 µm. **b** Distance of polarized MKs after local illumination was analyzed. Cells were preincubated with 10 µM CASIN or DMSO (-). n.t. indicates no transduction; XXM 2.0: ChR2 XXM2.0-expressing MKs. **c** Distance of polarized MKs of wild type and $Cdc42^{-/-}$ MKs was analyzed after local illumination. n.t. indicates no transduction, XXM 2.0 indicates expression of ChR2 XXM2.0. **d** Spread MKs on fibrinogen were incubated with 30 µM Rhosin or DMSO (-) before local illumination with blue light. Distance of polarized MK movement was determined. XXM 2.0: ChR2 XXM2.0-expressing MKs. **e** Distance of polarized movement of spread control, $Rac1^{-/-}$, $RhoA^{-/-}$ and $RhoB^{-/-}$ MKs on fibrinogen was determined. n.t. indicates no transduction; XXM 2.0: ChR2 XXM2.0-expressing MKs; WT: wild-type MKs. Results are mean ± s.d.

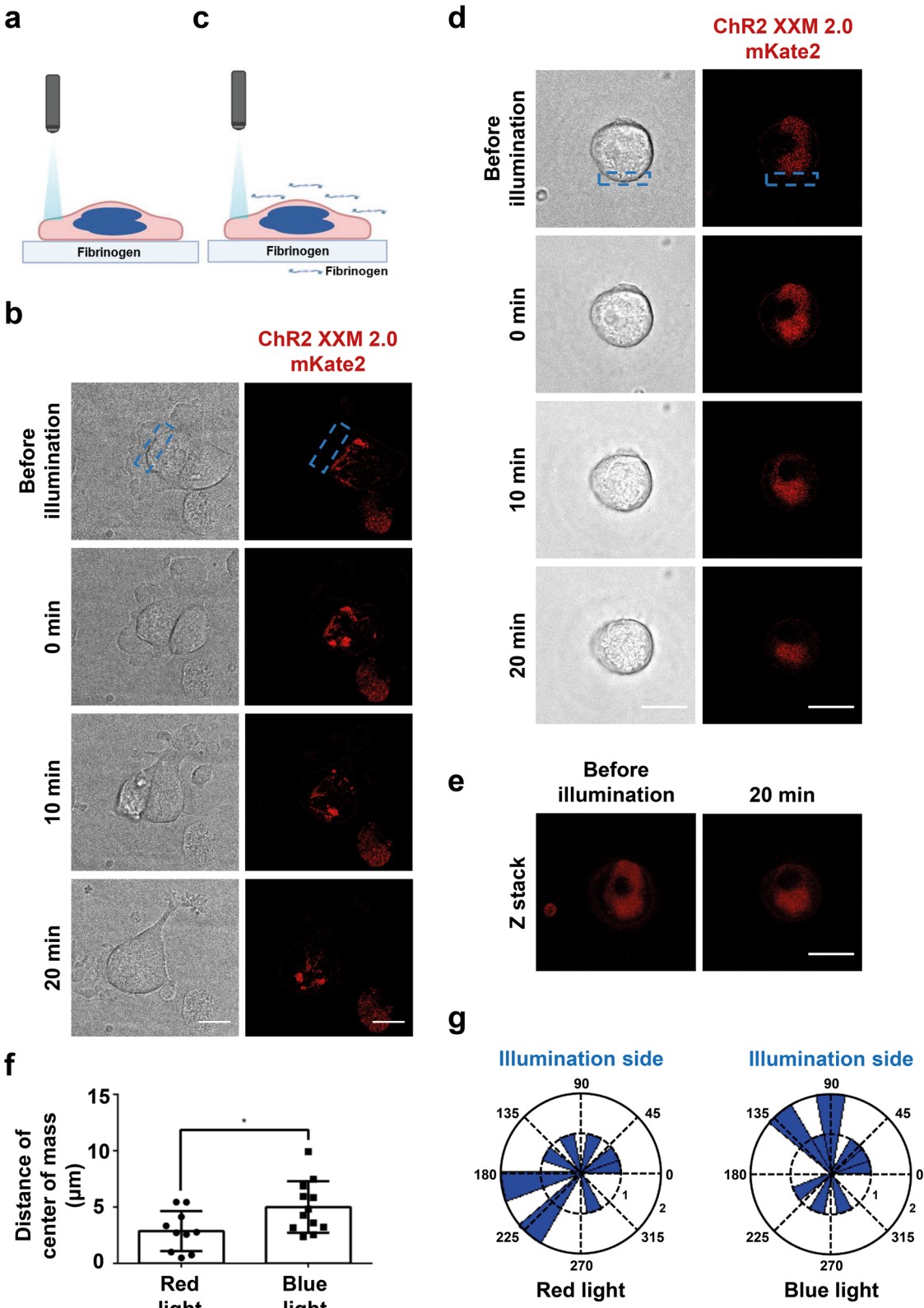

**Fig. 6 | Light-induced calcium influx triggers DMS polarization. a** Schematic of local illumination of ChR2 XXM2.0-mKate2 MKs. **b** Representative time-lapse images of ChR2 XXM2.0-mKate2 MKs under conditions of (**a**). ChR2 XXM2.0-mKate2 is expressed in the (demarcation) membrane. The blue dashed rectangle indicates illumination region. Scale bar indicates 22 μm. **c** Schematic of local illumination of ChR2 XXM2.0-mKate2 MKs in the presence of 100 μg/ml fibrinogen in buffer. **d** Representative time-lapse images of ChR2 XXM2.0-mKate2 MKs under conditions of (**c**). ChR2 XXM2.0-mKate2 is expressed in the (demarcation) membrane. The blue

dashed rectangle indicates illumination region. Scale bar indicates 22 μm. **e** Representative Z stack images of MK in (**d**) before and 20 min post illumination. Scale bar indicates 22 μm. **f** Quantification of polarized distance of center of mass of DMS in the experimental setting in (**d**). Red light (633 nm) that cannot activate ChR2 XXM2.0 was used as control. **g** Rose diagram illustrates DMS polarization direction at 20 min after 3 min local illumination in the experimental setting of (**d**, **f**). The radius of the black dashed ring indicates cell count. The black dashed lines indicate orientation. The schematic illustration was created with BioRender.com. Results are mean ± s.d.

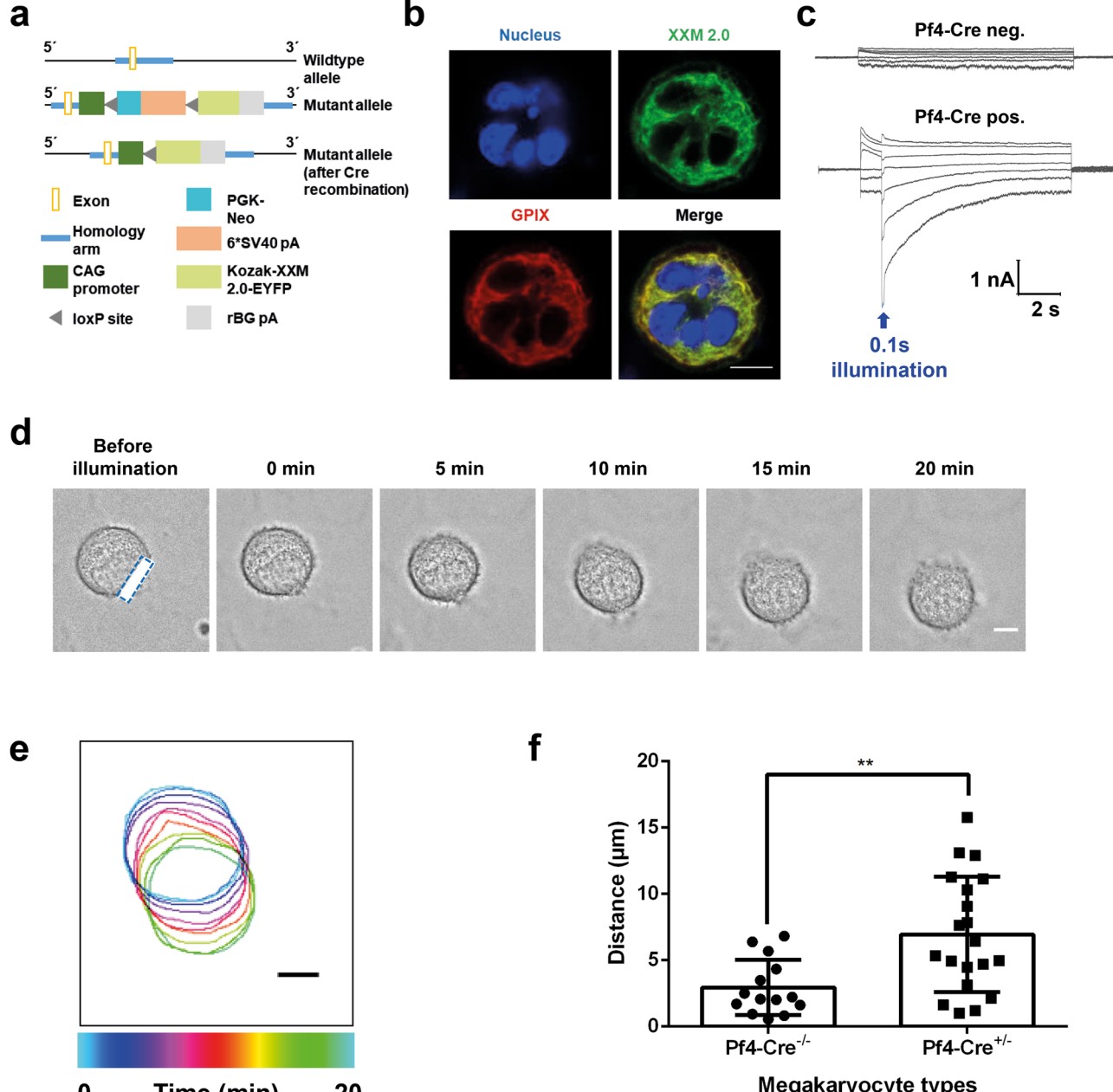

**Fig. 7 | Local illumination induces polarized movement of MKs from ChR2 XXM2.0-EYFP mice. a** Schematic representation of the targeting strategy. **b** Representative confocal images of ChR2 XXM2.0 MKs from the transgenic mouse line (loxP/Pf4-cre) stained for the nucleus (blue) with DAPI, GPIX (red) with Dylight 647 anti-GPIX antibody, and ChR2 XXM2.0 was visualized by its fusion protein EYFP (green). Scale bar indicates 10 μm. **c** Superimposed current traces for control MK (upper) and MK from the ChR2 XXM2.0 mouse line (loxP/Pf4-cre) (lower) after applying 0.1 s light pulse (473 nm, 550 μW/mm²) from a series of voltage pulses from −80 to +40 mV of 10 s duration with 20 mV increments from a holding potential of −40 mV. **d** Representative images of time-lapse microscopy of MK from the ChR2 XXM2.0 mouse line (loxP/Pf4-cre), which was locally illuminated for 1 min and then observed for 20 min. Local illumination was performed with FRAP module of a confocal microscope. The blue, dashed rectangle indicates illumination area. 0 min means time point directly after 3 min illumination. Scale bar indicates 10 μm. **e** Color-coded (time) representative cell outlines from the cell in (**d**). Scale bar indicates 10 μm. **f** Distance of polarized movement of ChR2 XXM2.0 expressing MKs (*Pf4-cre* +/-) and control MKs (*Pf4-cre* -/-) post local illumination. The polarization distance was analyzed by the distance between the center of mass of MKs before 1 min local illumination and after 20 min observation. Results are mean ± s.d.

bovine serum albumin density gradient was used on day 3 to separate MKs from non-MK cells. Experiments were performed on day 4.

### *Xenopus* oocyte expression and two-electrode voltage-clamp recording

cRNA-injected oocytes were incubated in ND96 solution (96 mM NaCl, 5 mM KCl, 1 mM MgCl₂, 1 mM CaCl₂, 5 mM HEPES, pH 7.4) with 1 μM all-trans-retinal at 16 °C. Two-electrode voltage-clamp (TEVC) recordings were performed 2 days after injection at room temperature.

For experiments with 80 mM Ca²⁺, 50 nl 200 mM of the fast Ca²⁺ chelator BAPTA (potassium salt) was injected into each oocyte (~10 mM final concentration in the oocyte). Injected oocytes were incubated for 1 h at 16 °C, and then the TEVC measurement was performed. A 473 nm laser (Changchun New Industries Optoelectronics Tech) was used as light source.

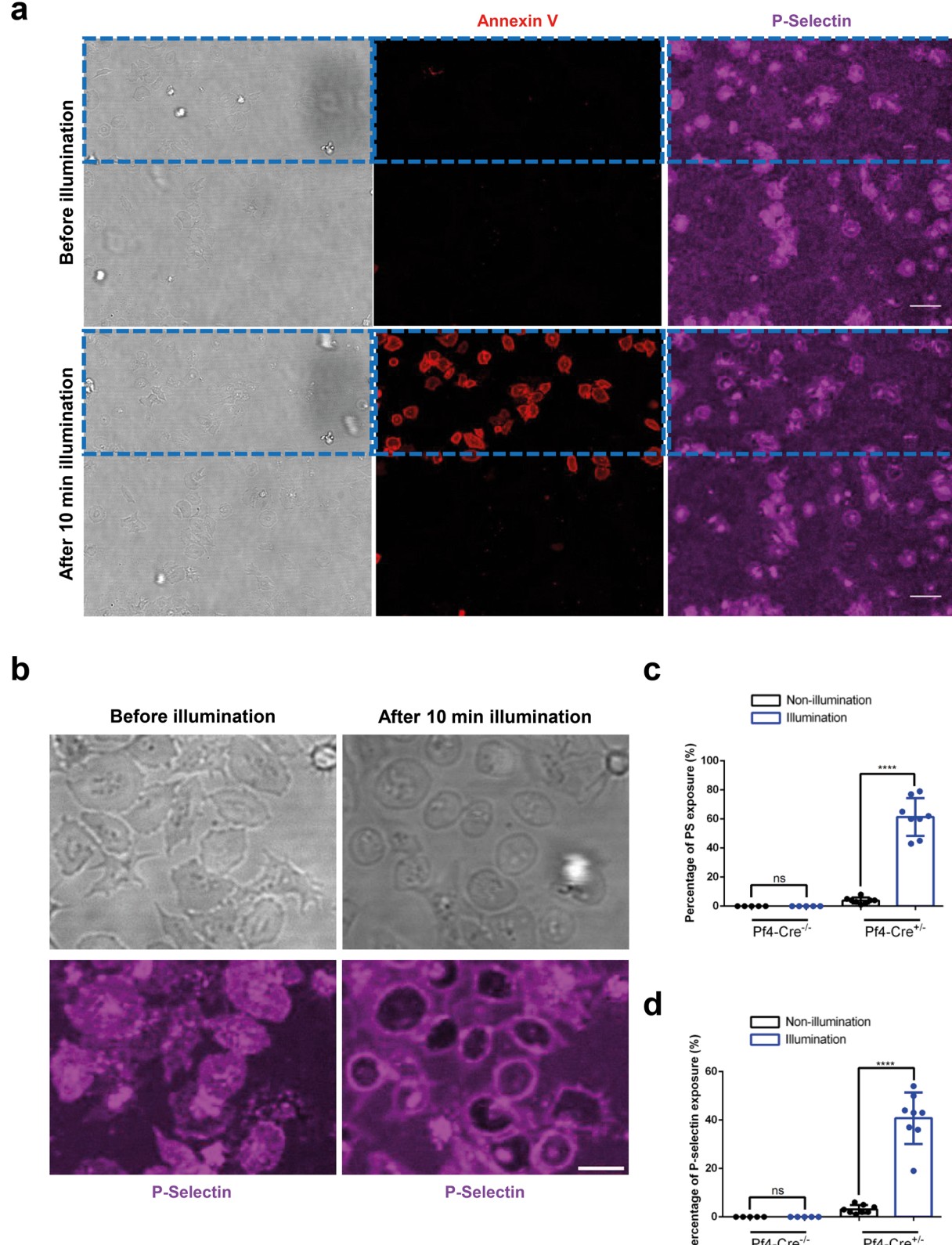

**Fig. 8 | Illumination induces PS and P-selectin exposure on platelets from transgenic mice. a** Platelets of control and homozygous (*tg/tg*) ChR2 XXM2.0-EYFP mice, spread on fibrinogen, were illuminated (as indicated with a dotted blue box) and PS (left, Annexin V positive) and P-selectin (right, anti-P-selectin) exposure were detected. Scale bar indicates 10 μm. **b** Spread platelets of homozygous (*tg/tg*) ChR2 XXM2.0-EYFP mice were illuminated and P-selectin (right, anti-P-selectin) exposure was detected. Scale bar indicates 5 μm. The percentage of platelet PS exposure (**c**) or P-selectin expression (**d**) after 10 min illumination. Results are mean ± s.d.

The light intensities were measured with a PLUS 2 Power & Energy Meter (LaserPoint s.r.l).

## Expression of constructs in megakaryocytes

Targeted DNAs in the MSCV vector were transfected into 293 T cells using the pCL vector system and ROTIFect from Roth. Viral supernatant was collected, and bone marrow-derived cells were infected on day 1.

## Immunofluorescence of megakaryocytes

To visualize protein expression and localization, cultured MKs were fixed and permeabilized with 4% PFA and 0.1% Triton-X-100 in PHEM-Buffer. The plasma membrane and demarcation membrane system were stained using Alexa 647-conjugated anti-GPIX antibody, and the nucleus with DAPI. Samples were visualized with a Leica TCS SP8 confocal microscope (Leica Microsystems).

## Whole-cell patch-clamp

Whole-cell currents were recorded from a single MK using a GeneClamp-500B amplifier (Axon Instruments) with an inverted microscope (Leica DMi8). Patch pipettes were prepared from borosilicate tubes (Kimble Chase) using a PC-10 vertical puller (Narishige) and polished with an MF2 microforge (Narishige). Experiments were performed at room temperature (~22 °C). The bath solution contained 145 mM NaCl, 5 mM KCl, 1 mM $CaCl_2$, 1 mM $MgCl_2$, 10 mM D-glucose, 10 mM HEPES, pH 7.35. The pipette solution contained 150 mM KCl, 0.1 mM EGTA, 1 mM $MgCl_2$, 0.05 mM $Na_2GTP$, 10 mM HEPES, pH 7.2. A 473 nm laser (Changchun New Industries Optoelectronics Tech) was used as a light source. Stimulation and data acquisition were controlled with NI USB-6221 (National Instruments) and WinWCP software (v4.1.7, 103 Strathclyde University, UK). Origin 2020 Pro was used for data analysis.

## Megakaryocyte spreading

Chamber slides (μ-Slide 8 Well, uncoated, Ibidi) were coated with 100 μg/ml fibrinogen or 50 μg/ml fibronectin at 4 °C overnight. After washing, chamber slides were blocked with 3% BSA/PBS for 1 h. Subsequently, MKs were seeded in chamber slides for 3 h at 37 °C and 5% $CO_2$. The chamber slides were washed, and 200 μl DMEM medium supplemented with 10% FCS, 1% penicillin/streptomycin was added.

## Illumination of megakaryocytes

Spread MKs were illuminated with the 488 nm laser line (10% laser power) of the Argon laser (22% laser power) using the FRAP module of a confocal laser scanning microscope (TCS SP8 confocal, Leica Microsystems) equipped with an HCPLAPO CS2 ×63/1.40 OIL objective. Region of interest covering peripheral parts of the MK (local illumination) or the entire MK (global illumination) was selected for blue light illumination to activate the ChR2 XXM2.0. HeNe 633 nm laser light (1% laser power) was used to observe MK behavior in brightfield. An area of 512 × 512 pixels was selected as the imaging region. The imaging interval was 30 s. Videos and images of MKs were generated with FIJI/IMAGEJ. MKs in suspension were illuminated with 100 μW/60 mm² LED light.

## Immunoblotting

Cell lysates were separated by SDS-PAGE and blotted onto polyvinylidene difluoride membranes. Blots were probed for MLC2 p-S19 (Sigma Aldrich, #SAB5700516, 1:1000), MLC2 (Cell Signalling, #8505S, 1:1000), or β-tubulin (Sigma Aldrich, T5293, 1:1000). Horseradish peroxidase-conjugated secondary antibodies and enhanced chemiluminescence solution (MoBiTec) were used for visualization. Immunoblots were recorded directly using an Amersham Imager 600 (GE Healthcare).

## MK polarization

The MK outline was manually drawn, and the center of mass was calculated by FIJI/ImageJ. The polarization distance was calculated by the distance between the two centers of mass of MKs before local illumination and 20 min observation after illumination. Color-coded time sequence of MK outlines was manually drawn by FIJI/ImageJ. MK polarization trajectory and the rose diagram were produced by Chemotaxis and Migration Tool 2.0.

## Platelet spreading and PS exposure

Slides were coated with human fibrinogen (100 μg/ml, Sigma), incubated at 4 °C overnight, and blocked with 1% BSA diluted in PBS for at least 30 min at room temperature. Subsequently, slides were washed with Tyrode's buffer with 2 mM $Ca^{2+}$. 150,000 platelets per μl in Tyrode's buffer with 2 mM $Ca^{2+}$, thrombin (f.c. 0.01 U/ml), 0.35 μg/ml Annexin V Alexa Fluor 546, and 18 μg/ml anti-P-selectin Alexa Fluor 647 antibody. The platelets were spread in ibidi dish for 30 min before 10 min illumination.

## Data analysis

Results are mean ± s.d.; individual data points were graphed in the figure. The statistical significance in the whole-cell patch-clamp experiments was analyzed using the Student's $T$-test. Differences between the two groups were analyzed using the Mann–Whitney two-tailed test. For more than two groups, the Kruskal–Wallis test followed by Dunn's test for multiple comparisons was performed using GraphPad Prism software. $P$-values < 0.05 were considered as statistically significant: $*P < 0.05$; $**P < 0.01$; $***P < 0.001$; $****P < 0.0001$. Results with a $P$-value > 0.05 were considered as not significant (ns in the figures).

## Data availability

The data generated and analyzed in this study are available from the corresponding authors on reasonable request. Source data for the graphs in the main and supplementary figures are provided in Supplementary Data 1. Original Western blot figures can be found in the supplement.

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

## Acknowledgements
We thank Bernhard Nieswandt and the microscopy platform of the Core Unit Imaging, Rudolf Virchow Center for Integrative and Translational Imaging for providing technical infrastructure and support, Daniela Naumann and Birgit Midloch for technical assistance, Xiao Duan for the help with oocyte injection, and Shang Yang and Chong Zhang for the cloning of some Channelrhodopsin variants. This work was supported by the Deutsche Forschungsgemeinschaft (DFG; German Research Foundation): Project numbers TR240 374031971 to M.B. and G.N., 452622672 to M.B. and 525167920 to M.B. and S.G.

## Author contributions
Performed experiments: Y.Z., J.Y.S., D.S., R.H., Z.N., S.G.; Data analysis: all authors; the ChR2 XXM2.0 transgene mouse was designed and ordered by C.X. and C.S.; Writing—original draft: Y.Z., S.G., M.B.; Supervised research: G.N., S.G., M.B.; Funding acquisition: M.B., G.N., S.G. All authors have critically revised and approved the final version of the manuscript.

## Funding

## Competing interests
The authors declare no competing interests.
