## [Transparent Peer Review file · Communications Biology]

Optogenetic Induction of Subcellular Ca²⁺ Events in Megakaryocytes and Platelets Using a Highly Ca²⁺-conductive Channelrhodopsin

Corresponding Author: Professor Markus Bender

Version 0:

Reviewer comments:

Reviewer #1

(Remarks to the Author)

This paper describes incorporation of a highly conductive calcium channelrhodopsin variant into bone marrow-derived megakaryocytes. Following functional characteristics, this approach was used to create optogenetic manipulation of anucleate blood platelets. This is a very concise submission with the following comments below:

1. Can the authors briefly describe the mutations, signal sequence and N terminal truncation which comprise ChR2 XXM2.0. This information does not appear to be present in the current manuscript. Perhaps, it would be helpful for readers to compare changes in sequences for each construct in Figure 1A. In addition, could the authors add information about why they did not compare photocurrents of the other calcium-conducting channels they mention in the Introduction in Figure 1A.
2. Did the authors normalize photocurrents in Figure 1A to surface expression? If not, this should be done to compare photocurrents between constructs.
3. Figure 1E does show the ChR2 variant on the plasma membrane, but there also appears to be significant intracellular localization of protein. It might be useful to mention this in the manuscript as light activation of intracellular ChR2 could have a cellular impact. Could the authors comment on this potential impact.
4. For Figure 2, can the authors provide a more detailed description of the differing panels. At the same time, outside of immunoblot analysis of MLC2 phosphorylation, is there a confirmatory molecular based assay that can be used to assess movement of MKs.
5. The authors have expressed the construct in mice. However, details of the results are very preliminary. It would greatly improve the manuscript if these details could be better described and/or a more targeted hypothesis-based question be asked/resolved through integration of this protein in MK function. As it is, the take home message appears to be that optogenetics may have potential for manipulating the function of specific platelet groups. In the discussion, the authors mention a variety of areas where a construct would be useful, most notably in the paragraph starting on line 246. But, none of these questions have been answered with this new mouse line.
6. This reviewer was a little confused by the text in the discussion. It says that deletion of NMDAR reduced platelet counts in mice. It doesn't seem like this work was done within this paper, nor was this analyzed here. It should be made more clear that this is from prior work and not the current submission.

Reviewer #2

(Remarks to the Author)

Zhang et al in this manuscript details the dynamics of Calcium signaling in MKs and platelets using the precision biological technique of optogenetics. The authors have devised a Calcium channelrhodopsin variant, ChR2 XXM2.0, which is an improvement over the existing variants of Calcium channelrhodopsin, to demonstrate the role of calcium in platelet and MK function. This construct and the subsequent transgenic mouse strain generated in this manuscript is novel and will serve as a beneficial tool for the platelet and MK field in general. The manuscript is well crafted and the potential of this technique, as an optogenetic toolbox, to answer diverse questions related to platelet production and function is very promising. The methods section is sufficiently detailed, and thus should be feasible to reproduce. The methods of statistical analysis are congruent with field standards.

The following are the major critiques, which if addressed, would benefit the manuscript.

1. In Fig1E, it appears that ChR2 XXM2.0 is expressed diffusely throughout the cytosol, with possible localization to DMS and other endosomal organelles. The authors should consider adding additional markers to better define the localization of this channelrhodopsin. Specifically, does ChR2 XXM2.0 also localize with intracellular calcium stores? The authors can use Stim1 or Orai1 as markers to address this.
2. In Fig1G, in their Cal 590 experiments, have the authors determined whether the extent of calcium influx induced by agonists (e.g., ADP, thrombin) is comparable to the calcium influx triggered by 3 min of light illumination in their XXM2.0 variant MKs? Furthermore, would combining both agonist- and light illumination- induced calcium influx lead to a synergistic effect in these MKs?
3. In Fig2B, the authors should show the center of mass of the MK in the diagram and illustrate the distance traveled by the MK after the illumination at each time point (using dotted lines or a similar approach). Without reviewing the corresponding Supplementary video, it is challenging to assess the directional movement of the MK based on the representative images shown in panel B. As it currently stands, panel B is somewhat confusing and does not add any meaningful information otherwise. Additionally, the authors should also consider adding a set of images in panel B with the BAPTA-AM data to show impaired polarized movement. This would complement the data and enhance the figure's impact.
4. As the authors have rightly stated, MK polarization towards the sinusoids precedes proplatelet formation. Have the authors examined whether the local illumination of XXM2.0 MKs, which drives directional movement, also coincides with the site of proplatelet formation? The authors can use MKs isolated from the PF4-Cre-ChR2 XXM2.0 mice and in vitro culturing them to see if local illumination induces proplatelet formation.
5. In Fig3, does adhesion on matrices like fibrinogen or fibronectin augment Myosin light chain phosphorylation? Have the authors investigated at downstream signaling pathways involved in MK adhesion and motility such as ERK1/2, Src-Syk-PLCgamma2, all of which have been implicated in MK migration and thrombopoiesis? Additionally, does BAPTA-AM chelation impair MLC phosphorylation in XXM2.0 MKs after light illumination?
6. The labels for the panels in Figure 4 panels are incorrect as there are two subpanels labeled "f". The Figure legend should be revised to reflect this correction.
7. In Fig4B and C, the authors may want to consider including the red fluorescent channel of the mKate2 ChR2 XXM2.0 MKs to show them as red. This could then be merged with the green channel of the Alexa 488-Fibrinogen, to highlight the overlap in yellow (or whichever color panel they deem to use), demonstrating the regions of α IIb β 3 binding. This will clarify the figure and better illustrate the point of localized binding and subsequent activation of α IIb β 3.
8. Can the authors explain their rationale to use JON/A Fab to block the binding site of activated α IIb β 3 rather than using eptifibatid?
9. Do the XXM2.0 MKs exhibit P-Selectin expression and PS exposure after light illumination, similar to the platelets from the transgenic mice?
10. In Fig6 and its associated figure legend, the authors should specify the matrix used for spreading the platelets from the transgenic mice.

Reviewer #3

(Remarks to the Author)

In this manuscript entitled "Optogenetic Induction of Subcellular Ca²⁺ Events in Megakaryocytes and Platelets Using a Highly Ca²⁺-conductive Channelrhodopsin", Zhang et al expressed the ChR2 XXM2.0 in primary bone marrow-derived MKs and found that blue light illumination induced a photocurrent and Ca²⁺ signals in those MKs. Local illumination of these MKs triggered polarized movement on fibrinogen, which was dependent on β 3 integrin-fibrinogen binding and myosin IIA activity. The authors also established a new transgenic mouse line, which enables expression of ChR2 XXM2.0 in different cell types when combined with available Cre lines, and found Ca²⁺-dependent phosphatidylserine and P-selectin exposure of spread platelets upon illumination. They claimed that optogenetic tool ChR2 XXM2.0 and the newly developed transgenic mouse line hold promise for further investigations into subcellular Ca²⁺ signaling dynamics across diverse cell types. The study is interesting and the submitted videos are useful. There is no major concern from this reviewer. However, the following questions and comments may be useful to further improve the quality of the manuscript.

- 1) The figure 1d legends for curves are unclear. If XXM2.0 will be used/highlighted for further experiments, the authors may use "red curve" to show it.
- 2) Figure 2b, an arrow (shows the direction) may be provided in the figures to help the readers for the "directional movement" of the MK cell.
- 3) Page 7, line 158, The authors may cite a review article regarding platelet integrins: Platelets in hemostasis and thrombosis: role of integrins and their ligands. PMID: 12725952
- 4) Page 7-8, in the section of "Local activation of ChR2 XXM2.0 triggers localized binding of integrin α IIb β 3 to fibrinogen", the authors elegantly studied the light-induced MK polarization, spread MKs on fibrinogen. As fibronectin is a key matrix protein and plasma fibronectin plays important roles in thrombosis and hemostasis (PMID: 12606706; PMID: 25180602; PMID: 20116835), likely important for MK polarization, spreading and migration. It will be interesting and important if the authors can examine whether the similar (or different) behavior can be observed in the n.t. and ChR2 XXM2.0 MKs on fibronectin. If the authors cannot perform the experiments, at least a discussion should be provided to readers.
- 5) Figure 6, although the conclusion seems to be clear in Figure 6cd, the figures are not easy for readers to understand the

P-selectin expression in figure 6ab. This can be improved.

Version 1:

Reviewer comments:

Reviewer #1

(Remarks to the Author)

This reviewer appreciates the response to the initial round of comments. These revisions certainly have improved the current draft. At the same time, a query from the initial round of review remains unresolved and can be further addressed.

1. In the initial comments, it was suggested that photocurrents should be normalized to photocurrents. Instead, the authors provide qualitative data on relative surface expression using fluorescence approaches. A more quantitative method would be to isolate surface expression of all proteins by biotinylating surface expressed proteins and then quantifying proteins following isolation with streptavidin beads. This data can then be used to normalize photocurrents. The current text describes "absolute Ca²⁺ currents". However, it is not clear what this means and could have a variety of interpretations.

Reviewer #2

(Remarks to the Author)

The authors have addressed all the concerns raised and the additional data generated has improved the manuscript.

Reviewer #3

(Remarks to the Author)

The authors have adequately addressed most of my questions/concerns.

However, as fibronectin is a key matrix protein and plasma fibronectin plays important roles in thrombosis and hemostasis (PMID: 12606706; PMID: 25180602) and fibronectin can self- and non-self assembly (PMID: 20116835, PMID: 27098513), it is likely also important for MK polarization, spreading and migration. If the authors "did not observe ChR2 XXM2.0 MK polarization after local illumination on a fibronectin-coated surface" (Page 14, line 8-9), it will be more convince if they can present the data and give a brief explanation for readers as this is an important finding and will certainly enrich us understanding this "integrin IIb 3-dependent manner on a fibrinogen (but not fibronectin)-coated surface" (Discussion, Page 14, line 7). If the authors cannot present their "unconclusive" or "preliminary data", a proper introduction/discussion including related references should be provided to the readers.

Version 2:

Reviewer comments:

Reviewer #1

(Remarks to the Author)

The authors have sufficiently addressed this reviewer's concerns.

Reviewer #3

(Remarks to the Author)

This is an interesting work, and the authors have adequately addressed my previous concerns and question. There is no more question from this reviewer.

However, the authors may underestimate the importance of their findings "we did not observe ChR2 XXM2.0 MK polarization following local illumination on a fibronectin-coated surface (Supplementary Fig. 9)." (Discussion, Page 14, line 12-14).

This reviewer recommends to highlight this discovery and replace the sentence "Fibronectin is also a ligand for IIb 3" (Discussion, page 14, line 12) with " Fibronectin is a key matrix protein and also a ligand of IIb 3, which plays important roles in IIb 3-medated thrombosis and hemostasis (cite references: PMID: 12855554; PMID: 12606706; PMID: 25180602)."

Response to the reviewers

We sincerely thank the reviewers for their positive and valuable comments on our manuscript. Below, we provide a detailed, point-by-point response. All changes in the manuscript are highlighted in red. We would like to mention that due to the revision work, we added two more co-authors to the author list.

Reviewer #1

This paper describes incorporation of a highly conductive calcium channelrhodopsin variant into bone marrow-derived megakaryocytes. Following functional characteristics, this approach was used to create optogenetic manipulation of anucleate blood platelets. This is a very concise submission with the following comments below.

1. Can the authors briefly describe the mutations, signal sequence and N terminal truncation which comprise ChR2 XXM2.0. This information does not appear to be present in the current manuscript. Perhaps, it would be helpful for readers to compare changes in sequences for each construct in Figure 1A. In addition, could the authors add information about why they did not compare photocurrents of the other calcium-conducting channels they mention in the Introduction in Figure 1A.

We appreciate the reviewer's insightful comment. To provide a clearer comparison, we have now included a schematic representation of the channelrhodopsins used in the study in the newly added Supplemental Figure 1a.

Regarding the choice of constructs for photocurrent comparison, we did not include CatCh and PsCatCh2.0 because: 1) they have already been compared to XXM in previous studies (Ref. 10); 2) PsCatCh2.0 offers advantages such as fast kinetics and high Ca^{2+} conductance but also exhibits higher Na^+ permeability and lower light sensitivity compared to XXM (Refs. 10, 11 and 12); Given that our study focuses on a cellular system that does not require fast responses typical of excitable cells, we opted for the XXM series to minimize light exposure - thereby reducing potential phototoxic effects - and to limit Na^+ -induced interference. We have added several clarifying sentences to the first paragraph of the Introduction section to make this rationale explicit.

"In our previous study, XXM was compared with CatCh and demonstrated a higher Ca^{2+} current. While PsCatCh 2.0 has advantages such as fast kinetics and high Ca^{2+} conductance, it also exhibits highly increased Na^+ permeability."

2. Did the authors normalize photocurrents in Figure 1A to surface expression? If not, this should be done to compare photocurrents between constructs.

We appreciate the reviewer's suggestion. To address this point, we have now included fluorescence images and quantification in the newly added Supplemental Figure 1b, c. As shown in the figure, XXM2.0 exhibits stronger expression compared to other constructs, a characteristic that can be attributed to the LR, T, and E sequence modifications. Additionally, we have added a sentence to the first paragraph in the Results section to clarify this point.

*"Our goal was to select a construct with both high Ca^{2+} permeability and light sensitivity. Following equal cRNA injection into *Xenopus* oocytes, XXM2.0 exhibited the highest expression among all constructs (Supplementary Fig. 1b, c), a characteristic attributed to its LR, T, and E sequence modifications. To compare their absolute Ca^{2+} currents, we measured the photocurrents in the bath solution containing 80 mM CaCl_2 ."*

However, due to the large size of oocytes and technical limitations, we are unable to precisely quantify plasma membrane-specific expression levels in this system. Nevertheless, in other cell types, XXM2.0 has demonstrated higher expression and stronger functional responses. This is further supported by independent validation from Dr. Andrew Hamilton (UC Davis), who recently evaluated XXM2.0 in *Xenopus laevis* tadpoles. In his feedback, Dr. Hamilton noted:

"All three of the ChR2 variants you sent (CapChR2, fCapChR2, and XXM2.0) have been very well tolerated by our X. laevis tadpoles. We've had good success invoking calcium influx with Cap and fCap, but even greater success with XXM2.0, which is so efficient at passing calcium that simply screening tadpoles injected with pCS2+ XXM2.0 for the integrated EYFP always invokes muscle flexion, even when the tadpoles are deep under anesthesia. It really is a very impressive construct."

3. Figure 1E does show the ChR2 variant on the plasma membrane, but there also appears to be significant intracellular localization of protein. It might be useful to mention this in the manuscript as light activation of intracellular ChR2 could have a cellular impact. Could the authors comment on this potential impact.

We thank the reviewer for this comment. The maturation of megakaryocytes includes the development of a unique and extensive membrane system known as the demarcation membrane system (DMS). These membrane invaginations do not separate from the surface membrane but remain continuous with it, creating a vast internal network. The DMS ensures sufficient membrane supply for the formation of thousands of platelets from a single megakaryocyte. Indeed, we believe that local or global illumination also activates the ChR2 in the invaginated membrane system.

We rephrased and added the following sentences in the result part:

"ChR2 XXM2.0, tagged with EYFP, co-localized with glycoprotein (GP) IX. This indicates that ChR2 XXM2.0 is expressed in both the plasma membrane and the internal demarcation membrane system (DMS) of megakaryocytes (MKs). The DMS provides a sufficient membrane supply for the formation of thousands of platelets from a single megakaryocyte."

We further analyzed how Ca^{2+} influx of ChR2 XXM2.0 in the outer and invaginated membrane region affects DMS polarization. In new figure 6 and Supplementary Video 3, we show:

"Local light activation of ChR2 XXM2.0 induces DMS polarization in MKs: DMS polarization is a prerequisite for directional proplatelet formation into sinusoidal vessels. Therefore, we investigated if light-induced Ca^{2+} influx also regulates DMS polarization in MKs. As ChR2 XXM2.0 also localizes in the DMS, we utilized the red fluorescence tag mKate2 of ChR2 XXM2.0-mKate2 as a marker to observe DMS arrangement and to avoid activation of ChR2 XXM2.0 during the observation period (Fig. 6a). We found ChR2 XXM2.0-mKate2, indicative for the DMS, polarized towards the leading edge of polarized MKs (Fig. 6a-b; Supplementary Video 3). In order to distinguish the DMS polarization from the cell movement, spread MKs were incubated with fibrinogen in suspension before local illumination to block polarized MK movement (Fig. 6c). While the polarized movement of ChR2 XXM2.0-mKate2 MK was prevented, the DMS still polarized towards the illumination area (Fig. 6d-g; Supplementary Video 3). This observation indicates that restructuring of the internal DMS is triggered by local Ca^{2+} influx."

4. For Figure 2, can the authors provide a more detailed description of the differing panels. At the same time, outside of immunoblot analysis of MLC2 phosphorylation, is there a confirmatory molecular based assay that can be used to assess movement of MKs.

As requested, we now provide more detailed description of the differing panels for figure 2 in the figure legend.

We thank the reviewer for this interesting question about other confirmatory molecular data. The GTPase Cdc42 was reported to be involved in megakaryocyte polarization and together with the GTPase RhoA are molecular checkpoints, which control transendothelial platelet biogenesis (PMID: 28643773; PMID: 26991240). We tested in new figure 5 the role of the different GTPases in light-induced megakaryocyte polarization and found that the small Rho GTPase Cdc42 is crucial for light-induced MK polarization. We added the following text in the result part:

“The small Rho GTPase Cdc42 is crucial for light-induced MK polarization:

In cells other than MKs, previous studies have shown that integrin binding to extracellular matrix proteins triggers selective activation of Rho GTPases, such as Cdc42, which induces cell polarization and migration. Importantly, Cdc42 was reported to be involved in MK polarization and together with RhoA are molecular checkpoints, which control transendothelial platelet biogenesis. To test the role of the small Rho GTPases in polarized MK movement, we used an inhibitor-based approach and genetic knockout mice. Our data show that cell polarization after local illumination of ChR2 XXM2.0 MKs is dependent on the small GTPase Cdc42, based on experiments using the Cdc42-specific inhibitor CASIN (Fig. 5a,b) and Cdc42-deficient MKs (Fig. 5c). We next studied the role of other small GTPases, such as RhoA, RhoB, and Rac1, in light-induced ChR2 XXM2.0 MK polarization. After preincubation with 30 μ M of the Rho inhibitor Rhosin, ChR2 XXM2.0 MKs did not show impaired polarization (Fig. 5d), indicating RhoA and RhoB are not involved in light-induced ChR2 XXM2.0 MK polarization. Consistently, there was also no difference in light-induced ChR2 XXM2.0 MK polarization between wild-type MKs and RhoA^{-/-} or RhoB^{-/-} MKs (Fig. 5e). Furthermore, Rac1^{-/-} MKs exhibited similar light-induced ChR2 XXM2.0 MK polarization compared to wild-type MKs (Fig. 5e). Taken together, we demonstrated that Cdc42, but not RhoA, RhoB, and Rac1, is crucial for light-induced ChR2 XXM2.0 MK polarization and motility mediated by Ca²⁺.”

We also discussed our new data and added the following text to the discussion part:

“A central role in establishing cell polarity has been demonstrated for the small GTPase Cdc42 of the Rho family. During the last years, experimental evidence has accumulated suggesting that Cdc42 activity in MKs is associated with polarized DMS formation and transendothelial platelet biogenesis. Our results support these findings because Cdc42 deficiency or inhibition of its activity prevented MKs from polarized movement. However, the detailed molecular mechanism how Cdc42 is involved in this light-induced process has not been addressed. It was reported that the Ca²⁺-dependent interaction of Lis1 with IQGAP1 promotes Cdc42 activation and induces neuronal motility. Another study demonstrated that the interaction of integrins with the extracellular matrix at the newly formed cell front leads to the activation and polarized recruitment of Cdc42 in astrocytes.”

5. The authors have expressed the construct in mice. However, details of the results are very preliminary. It would greatly improve the manuscript if these details could be better described and/or a more targeted hypothesis-based question be asked/resolved through integration of this protein in MK function. As it is, the take home message appears to be that optogenetics may have potential for manipulating the function of specific platelet groups. In the discussion, the authors mention a variety of areas where a construct would be useful, most notably in the paragraph starting on line 246. But, none of these questions have been answered with this new mouse line.

The reviewer raises an interesting point. We began this project in 2018 by expressing the optogenetic construct in megakaryocytes via viral infection. The focus of this manuscript is on ChR2 XRM2.0 and the in vitro analysis of megakaryocyte function. Based on our promising results, we decided to generate a mouse line to study megakaryocytes in vivo and examine platelet function. Optogenetic manipulation of megakaryocytes in vivo requires two-photon microscopy combined with a specific setup, which we are currently establishing. To strengthen this manuscript, we included preliminary data from the transgenic mice to (I) validate our MK findings with the viral approach and (II) to demonstrate for the first time that optogenetic manipulation of platelet function is possible. However, we have just started to analyze the platelet phenotype. Therefore, our platelet analyses are ongoing, and we plan to publish these results in a separate article.

6. This reviewer was a little confused by the text in the discussion. It says that deletion of NMDAR reduced platelet counts in mice. It doesn't seem like this work was done within this paper, nor was this analyzed here. It should be made more clear that this is from prior work and not the current submission.

As requested, we rephrased the sentence referring to reference 18 in the discussion part to avoid confusion:

"In support of these results, studies unveiled that deletion of the glutamate-gated N-methyl-D-aspartate receptor (NMDAR) with high Ca²⁺ permeability in MKs results in reduced platelet counts in mice, and inhibition of extracellular Ca²⁺ inflow also affected MK interaction with extracellular matrix proteins."

Reviewer #2

Zhang et al in this manuscript details the dynamics of Calcium signaling in MKs and platelets using the precision biological technique of optogenetics. The authors have devised a Calcium channelrhodopsin variant, ChR2 XRM2.0, which is an improvement over the existing variants of Calcium channelrhodopsin, to demonstrate the role of calcium in platelet and MK function. This construct and the subsequent transgenic mouse strain generated in this manuscript is novel and will serve as a beneficial tool for the platelet and MK field in general. The manuscript is well crafted and the potential of this technique, as an optogenetic toolbox, to answer diverse questions related to platelet production and function is very promising. The methods section is sufficiently detailed, and thus should be feasible to reproduce. The methods of statistical analysis are congruent with field standards. The following are the major critiques, which if addressed, would benefit the manuscript.

1. In Fig1E, it appears that ChR2 XRM2.0 is expressed diffusely throughout the cytosol, with possible localization to DMS and other endosomal organelles. The authors should consider adding additional markers to better define the localization of this channelrhodopsin. Specifically, does ChR2 XRM2.0 also localize with intracellular calcium stores? The authors can use Stim1 or Orai1 as markers to address this.

We co-stained megakaryocytes with antibodies directed against Stim1, Orai1, the early endosome marker EEA1 and the lysosomal marker Lamp1. We added the new data as new Supplemental Figure 2. We show that ChR2 XRM2.0, not only co-localizes with GPIIb/IIIa, but also partially co-localizes with Orai1 at the plasma membrane and we detected ChR2 XRM2.0 to

be in close proximity to Stim1. In contrast, the early endosomal marker EEA1 and lysosomal marker Lamp1 showed no comparable staining pattern with ChR2 XXM2.0.

2. In Fig1G, in their Cal 590 experiments, have the authors determined whether the extent of calcium influx induced by agonists (e.g., ADP, thrombin) is comparable to the calcium influx triggered by 3 min of light illumination in their XXM2.0 variant MKs? Furthermore, would combining both agonist- and light illumination- induced calcium influx lead to a synergistic effect in these MKs?

As requested, we performed this experiment and observed that 100 μ M ADP or 0.1 u/mL thrombin did not notably increase intracellular calcium levels, whereas CRP (10 μ g/mL) produced a short calcium spike. Since it is difficult to directly compare the different methods, we decided to insert the figure only in the response letter for the reviewer.

Legend to figure 1 to the reviewer: (a) Representative line plots of Cal590 fluorescence intensity in ChR2XXM2.0 positive megakaryocytes before and after stimulation with either 100 μ M ADP, 0.1 U/ml thrombin or 10 μ g/ml CRP. The sudden absence of fluorescence intensity indicates the time point when the respective agonist was added to the imaging chamber, for which detection had to be shut off. Fluorescence intensity remained virtually unchanged by ADP and thrombin, whereas CRP induced a short (>10 s) increase of fluorescence signal. (b) Time series of a representative ChR2XXM2.0 positive megakaryocyte stimulated with CRP displaying a slight increase of Cal590 fluorescence ~190 s after addition of CRP to the imaging chamber.

3. In Fig2B, the authors should show the center of mass of the MK in the diagram and illustrate the distance traveled by the MK after the illumination at each time point (using dotted lines or a similar approach). Without reviewing the corresponding Supplementary video, it is challenging to assess the directional movement of the MK based on the representative images shown in panel B. As it currently stands, panel B is somewhat confusing and does not add any meaningful information otherwise. Additionally, the authors should also consider adding a set of images in panel B with the BAPTA-AM data to show impaired polarized movement. This would complement the data and enhance the figure's impact.

As requested, we have incorporated the trajectory of the center of mass of the megakaryocyte from Figure 2b into Figure 2d. Additionally, we have rearranged Figure 2b and added the arrow to indicate the directional movement of the megakaryocyte. Furthermore, we have included images of megakaryocytes treated with BAPTA-AM after local illumination in Figure 2h.

4. As the authors have rightly stated, MK polarization towards the sinusoids precedes proplatelet formation. Have the authors examined whether the local illumination of XXM2.0 MKs, which drives directional movement, also coincides with the site of proplatelet formation? The authors can use MKs isolated from the PF4-Cre-ChR2 XXM2.0 mice and in vitro culturing them to see if local illumination induces proplatelet formation.

This is a very interesting question, which we have already tried to address. In the publication (PMID: 24463213, Di Buduo et al 2014 Haematologica) of the Balduini lab, the authors demonstrated that ADP promotes Ca^{2+} release from intracellular stores which is responsible for the regulation of proplatelet formation and for the activation of Store-Operated Calcium Entry, which promotes MK adhesion and migration. In our experimental set-up, focal illumination (calcium entry) induced polarized MK movement on fibrinogen. To avoid MK movement in an uncoated cell culture plate, we added MKs in vitro to low percentage of methylcellulose and focally illuminated the cells. However, we could also not induce proplatelet formation under these conditions. One option for the future might be to use a more ER targeted ChR2 construct.

Below, data of MKs embedded with methylcellulose followed by local illumination, as figure 2 to the reviewer. The lack of polarized movement demonstrates that the process is dependent on receptor – ECM protein interaction.

5. In Fig3, does adhesion on matrices like fibrinogen or fibronectin augment Myosin light chain phosphorylation? Have the authors investigated at downstream signaling pathways involved in MK adhesion and motility such as ERK1/2, Src-Syk-PLCgamma2, all of which have been implicated in MK migration and thrombopoiesis? Additionally, does BAPTA-AM chelation impair MLC phosphorylation in XXM2.0 MKs after light illumination?

The detection of phosphorylation of MLC-2 after illumination was performed in suspension to have enough material for the experiment.

Unfortunately, we could not induce polarized MK movement on fibronectin, which we now mention in the discussion part.

This is an interesting point. We performed immunoblot analysis to assess the phosphorylation of SFK, Akt, and Erk1/2 following global illumination. Our results show increased phosphorylation of SFK at Y418, Akt at S473, and Erk1/2 at T202/Y204 after 3 minutes of global illumination in ChR2 XXM2.0-expressing MKs in new Supplementary Figure 6. Since MLC phosphorylation is known to be Ca^{2+} -dependent, it is reasonable to expect that BAPTA-AM would inhibit this process. However, we did not perform this specific experiment. Future studies could explore this aspect in more mechanistic detail.

6. The labels for the panels in Figure 4 panels are incorrect as there are two subpanels labeled “f”. The Figure legend should be revised to reflect this correction.

We corrected the labeling.

7. In Fig4B and C, the authors may want to consider including the red fluorescent channel of the mKate2 ChR2 XXM2.0 MKs to show them as red. This could then be merged with the green channel of the Alexa 488-Fibrinogen, to highlight the overlap in yellow (or whichever color panel they deem to use), demonstrating the regions of α IIb β 3 binding. This will clarify the figure and better illustrate the point of localized binding and subsequent activation of α IIb β 3.

We appreciate the reviewer’s suggestion. After confirming that the MKs expressed mKate2-ChR2 XXM2.0, we did not capture images of mKate2-ChR2 XXM2.0 during the observation of Alexa 488-Fibrinogen binding. Instead, we now provide in figure 4 bright-field images of MKs at different time points to illustrate α IIb β 3 binding to the plasma membrane.

8. Can the authors explain their rationale to use JON/A Fab to block the binding site of activated α IIb β 3 rather than using eptifibatide?

According to the study of Magallon J et al., *Circulation*, 2011 (PMID: 21220740), eptifibatide is less effective at blocking platelet aggregation in mice compared to humans. This reduced efficacy is attributed to species-specific differences in the integrin α IIb β 3 receptor between mice and humans, which impacts the binding affinity of eptifibatide.

It is well described that JON/A binds to and functionally blocks murine α IIb β 3 (Bergmeier et al. 2002 <https://doi.org/10.1002/cyto.10114>).

9. Do the XXM2.0 MKs exhibit P-Selectin expression and PS exposure after light illumination, similar to the platelets from the transgenic mice?

We thank the reviewer for this question. We show in our new Supplementary Figure 4 that global illumination induces PS exposure on ChR2 XXM2.0 MKs. We performed 90 min global illumination of ChR2 XXM2.0 MKs with LED light (488 nm, 100 μ W/60 mm²). Annexin V was used to stain phosphatidylserine. We show that PS exposure on MKs starts between 10 and 30 minutes after illumination. Interestingly, we demonstrate that a higher percentage of MKs expressing ChR2 XXM2.0 with enhanced Ca²⁺ permeability expose PS than ChR2 H134R MKs.

Accordingly, we added the following text in the result part:

“Long-term (90 min) global illumination of ChR2 XXM2.0 MKs resulted in phosphatidylserine (PS) exposure on the plasma membrane (Supplementary Fig. 4a,b). In contrast, the PS exposure percentage of ChR2 H134R MKs was reduced compared to ChR2 XXM2.0 MKs (~20% vs ~60%) (Supplementary Fig. 4c). Given the possibility of phototoxicity of the fluorescence protein, we only expressed YFP in MKs and globally illuminated MKs for 90 min. YFP-expressing MKs did not show an increased percentage of PS exposure (Supplementary Fig. 4d). This finding suggests that long-term channel opening is detrimental to cell.”

10. In Fig6 and its associated figure legend, the authors should specify the matrix used for spreading the platelets from the transgenic mice.

Platelets were spread on fibrinogen. As requested, this information was added to the result part and to the figure legend. This figure is now figure 8.

Reviewer #3:

In this manuscript entitled “Optogenetic Induction of Subcellular Ca²⁺ Events in Megakaryocytes and Platelets Using a Highly Ca²⁺-conductive Channelrhodopsin”, Zhang et al expressed the ChR2 XXM2.0 in primary bone marrow-derived MKs and found that blue light illumination induced a photocurrent and Ca²⁺ signals in those MKs. Local illumination of these MKs triggered polarized movement on fibrinogen, which was dependent on beta3 integrin-fibrinogen binding and myosin IIA activity. The authors also established a new transgenic mouse line, which enables expression of ChR2 XXM2.0 in different cell types when combined with available Cre lines, and found Ca²⁺-dependent phosphatidylserine and P-selectin exposure of spread platelets upon illumination. They claimed that optogenetic tool ChR2 XXM2.0 and the newly developed transgenic mouse line hold promise for further investigations into subcellular Ca²⁺ signaling dynamics across diverse cell types. The study is interesting

and the submitted videos are useful. There is no major concern from this reviewer. However, the following questions and comments may be useful to further improve the quality of the manuscript.

1) The figure 1d legends for curves are unclear. If XXM2.0 will be used/highlighted for further experiments, the authors may use "red curve" to show it.

We appreciate the reviewer's suggestion. To address this point, we have now changed the colors of the curves accordingly.

2) Figure 2b, an arrow (shows the direction) may be provided in the figures to help the readers for the "directional movement" of the MK cell.

As request, we added an arrow to indicate the directional movement.

3) Page 7, line 158, The authors may cite a review article regarding platelet integrins: Platelets in hemostasis and thrombosis: role of integrins and their ligands. PMID: 12725952

As requested, we added the review article about platelet integrins as new reference 28.

4) Page 7-8, in the section of "Local activation of ChR2 XXM2.0 triggers localized binding of integrin α IIb β 3 to fibrinogen", the authors elegantly studied the light-induced MK polarization, spread MKs on fibrinogen. As fibronectin is a key matrix protein and plasma fibronectin plays important roles in thrombosis and hemostasis (PMID: 12606706; PMID: 25180602; PMID: 20116835), likely important for MK polarization, spreading and migration. It will be interesting and important if the authors can examine whether the similar (or different) behavior can be observed in the n.t. and ChR2 XXM2.0 MKs on fibronectin. If the authors cannot perform the experiments, at least a discussion should be provided to readers.

We thank the reviewer for this interesting question. We have performed the same experiment with a fibronectin coated surface, and we did not observe ChR2 XXM2.0 MK polarization after 3 min local illumination followed by 20 min observation. We mention this now in the discussion part.

"However, we did not observe ChR2 XXM2.0 MK polarization after local illumination on a fibronectin-coated surface."

5) Figure 6, although the conclusion seems to be clear in Figure 6cd, the figures are not easy for readers to understand the P-selectin expression in figure 6ab. This can be improved.

As requested, we added blue boxes to all images in Figure 6a, which is now Figure 8a, to indicate illumination area. In addition, we replaced Figure 6b, which is now Figure 8b, with more clear images for platelet ballooning and P-selectin exposure.

Response to the referees:

We sincerely thank the reviewers for their positive and valuable comments on our manuscript. Below, we provide a detailed, point-by-point response. All changes in the manuscript are highlighted in red. We would like to mention that due to the revision work, we added two more co-authors to the author list.

Reviewer #1 (Remarks to the Author):

This reviewer appreciates the response to the initial round of comments. These revisions certainly have improved the current draft. At the same time, a query from the initial round of review remains unresolved and can be further addressed.

1. In the initial comments, it was suggested that photocurrents should be normalized to photocurrents. Instead, the authors provide qualitative data on relative surface expression using fluorescence approaches. A more quantitative method would be to isolate surface expression of all proteins by biotinylating surface expressed proteins and then quantifying proteins following isolation with streptavidin beads. This data can then be used to normalize photocurrents. The current text describes “absolute Ca²⁺ currents”. However, it is not clear what this means and could have a variety of interpretations.

Response: We appreciate the reviewer’s thoughtful suggestion regarding the normalization of photocurrents to surface expression levels. We fully agree that this is an important factor for understanding the biophysical properties of channelrhodopsins. However, in the context of optogenetic applications, total photocurrent amplitudes are often more relevant and commonly used than normalized values, as the biological effect, such as downstream signaling or physiological response, depends on total ion flux, rather than current density per unit of protein.

While we prefer to present total photocurrents to reflect functional performance, we agree with the reviewer that expression levels should be addressed. We would also like to emphasize that enhanced expression is an intrinsic feature of ChR2 X2M2.0, and it contributed partially to the larger photocurrent. Thus we have included quantitative fluorescence measurements of protein expression. Importantly, we want to point out that these fluorescence values in Supplementary figure 1b were not obtained from microscopy images (Supplementary figure 1b), but from isolated membrane fractions measured using a fluorometer. This method has been successfully used in previous studies to estimate membrane-targeted expression of rhodopsins and other proteins (see: Scheib et al., *Nat Commun* 9, 2046 (2018), <https://doi.org/10.1038/s41467-018-04428-w>, and Yang et al., *BMC Biol* 19, 227 (2021), <https://doi.org/10.1186/s12915-021-01151-9>).

Regarding the reviewer’s suggestion to use surface biotinylation, we agree that it is a valuable technique for assessing membrane localization of a single protein under different conditions. However, it is less suitable for quantitative comparisons between different proteins, particularly when they vary in topology, extracellular loop structure, and lysine residue content, as is the case for the ChR variants studied here. These differences can significantly affect biotinylation efficiency. Additionally, the multi-step process involved (e.g., homogenization, washing, elution) introduces

variability and risk of differential protein loss for different proteins, further limiting the accuracy of direct comparisons.

While we maintain our focus on total photocurrents, we recognize the reviewer's concern and have revised the manuscript to better address this point. Specifically, we added the following clarification to the Results section:

“Notably, ChR2 X XM2.0 had the most significant Ca^{2+} current compared with ChR2 H134R⁴, X XM¹⁰ and the recently published CapChR2^{12,13}, particularly under lower light intensities (Fig. 1a, b). **This enhanced photocurrent likely results from a combination of increased expression levels (Supplementary Fig. S1), higher Ca^{2+} selectivity¹⁶, and potentially larger ion conductance.**”

[new text marked in red]

We also agree with the reviewer that the term "absolute Ca^{2+} current" was not clearly explained. We have revised the terminology related to "absolute Ca^{2+} current" to improve clarity. Our original intention was to describe photocurrents measured under high external Ca^{2+} conditions, where the inward current is predominantly due to Ca^{2+} influx. To avoid ambiguity, we have changed the text to:

“To compare their **Ca^{2+} currents**, we measured the photocurrents in the bath solution containing 80 mM CaCl_2 , **where the inward current is predominantly carried by Ca^{2+} .**”

[modified text marked in red]

Reviewer #2 (Remarks to the Author):

The authors have addressed all the concerns raised and the additional data generated has improved the manuscript.

Reviewer #3 (Remarks to the Author):

The authors have adequately addressed most of my questions/concerns.

However, as fibronectin is a key matrix protein and plasma fibronectin plays important roles in thrombosis and hemostasis (PMID: 12606706; PMID: 25180602) and fibronectin can self- and none-self assembly (PMID: 20116835, PMID: 27098513), it is likely also important for MK polarization, spreading and migration. If the authors “did not observe ChR2 X XM2.0 MK polarization after local illumination on a fibronectin-coated surface” (Page 14, line 8-9), it will be more convince if they can present the data and give a brief explanation for readers as this is an important finding and will certainly enrich us understanding this “integrin α IIb β 3-dependent manner on a fibrinogen (but not fibronectin)-coated surface” (Discussion, Page 14, line 7). If the

authors cannot present their “unconclusive” or “preliminary data”, a proper introduction/discussion including related references should be provided to the readers.

Response: We thank the reviewer for this comment and fully agree that the data should be shown. We analyzed additional cells under the experimental conditions using a fibronectin-coated surface, and the results are now presented in the new Supplementary Figure 9. We did not observe polarized megakaryocyte movement on the fibronectin-coated surface. One possible explanation for the differing results between fibrinogen- and fibronectin-coated surfaces is the difference in their binding sites for the integrin α IIb β 3. We have added the following sentences to the discussion section:

Fibronectin is also a ligand for α IIb β 3. However, we did not observe ChR2 XXM2.0 MK polarization following local illumination on a fibronectin-coated surface (Supplementary Fig. 9). One possible explanation is that fibronectin can bind to α IIb β 3 but primarily engages α 5 β 1, whereas fibrinogen is the main physiological ligand of α IIb β 3 and provides multiple high-affinity, multivalent binding motifs, such as the γ -chain HHLGGAKQAGDV sequence, which is absent in fibronectin (reference 28 Ni, H. & Freedman, J. Transfus Apher Sci., 2003).

Response to the referees:

We sincerely thank the reviewers for their positive and valuable comments on our manuscript. Below, we provide a detailed, point-by-point response. All changes in the manuscript are highlighted in red.

Reviewer #1 (Remarks to the Author):

The authors have sufficiently addressed this reviewer's concerns.

Reviewer #3 (Remarks to the Author):

This is an interesting work, and the authors have adequately addressed my previous concerns and question. There is no more question from this reviewer.

However, the authors may underestimate the importance of their findings "we did not observe ChR2 X XM2.0 MK polarization following local illumination on a fibronectin-coated surface (Supplementary Fig. 9)." (Discussion, Page 14, line 12-14).

This reviewer recommends to highlight this discovery and replace the sentence "Fibronectin is also a ligand for α IIb β 3" (Discussion, page 14, line 12) with " Fibronectin is a key matrix protein and also a ligand of α IIb β 3, which plays important roles in α IIb β 3-mediated thrombosis and hemostasis (cite references: PMID: 12855554; PMID: 12606706; PMID: 25180602)."

Response: We thank the reviewer 3 for this constructive comment and for highlighting the importance of our observation regarding the lack of ChR2 X XM2.0 MK polarization on a fibronectin-coated surface. We agree that the difference between fibrinogen and fibronectin in this context is both interesting and significant. At the same time, we recognize that this finding requires further investigation to fully understand the underlying mechanisms.

Following the reviewer's suggestion, we have revised the sentence "Fibronectin is also a ligand for α IIb β 3" to:

"Fibronectin is also a key matrix protein and a ligand of α IIb β 3, which was reported to be important in α IIb β 3-mediated thrombosis and hemostasis."

In addition, we now state: “The difference between fibrinogen- and fibronectin-dependent responses that we observed here is intriguing and deserves further in-depth investigation. One possible explanation might be (...)”

We believe these changes appropriately emphasize the relevance of our finding while making clear that additional studies will be needed to clarify the mechanistic basis of the differential responses to fibrinogen and fibronectin.